# Autonomous Purkinje cell activation instructs bidirectional motor learning through evoked dendritic calcium signaling

Audrey Bonnan[1], Matthew M. J. Rowan [1,2], Christopher A. Baker [1], M. McLean Bolton[1] & Jason M. Christie [1,3✉]

The signals in cerebellar Purkinje cells sufficient to instruct motor learning have not been systematically determined. Therefore, we applied optogenetics in mice to autonomously excite Purkinje cells and measured the effect of this activity on plasticity induction and adaptive behavior. Ex vivo, excitation of channelrhodopsin-2-expressing Purkinje cells elicits dendritic $Ca^{2+}$ transients with high-intensity stimuli initiating dendritic spiking that additionally contributes to the $Ca^{2+}$ response. Channelrhodopsin-2-evoked $Ca^{2+}$ transients potentiate co-active parallel fiber synapses; depression occurs when $Ca^{2+}$ responses were enhanced by dendritic spiking. In vivo, optogenetic Purkinje cell activation drives an adaptive decrease in vestibulo-ocular reflex gain when vestibular stimuli are paired with relatively small-magnitude Purkinje cell $Ca^{2+}$ responses. In contrast, pairing with large-magnitude $Ca^{2+}$ responses increases vestibulo-ocular reflex gain. Optogenetically induced plasticity and motor adaptation are dependent on endocannabinoid signaling, indicating engagement of this pathway downstream of Purkinje cell $Ca^{2+}$ elevation. Our results establish a causal relationship among Purkinje cell $Ca^{2+}$ signal size, opposite-polarity plasticity induction, and bidirectional motor learning.

[1] Max Planck Florida Institute for Neuroscience, Jupiter, FL, USA. [2] Present address: Emory University School of Medicine, Atlanta, GA, USA. [3] Present address: University of Colorado School of Medicine, Aurora, CO, USA. ✉email: jason.m.christie@cuanschutz.edu

Animals learn to adapt their movements in response to motor errors. Corrective learning may necessitate the strengthening and/or weakening of movements to restore mistake-free performance on subsequent trials. Therefore, neural circuits for motor learning must be able to flexibly signal and respond to different erroneous contexts that ultimately accommodate a range of behavioral modifications[1]. In the cerebellar cortex, error-driven motor learning entails alterations in the sensorimotor-evoked simple spiking pattern of Purkinje cells (PCs) to improve movement[2,3]. Because parallel fiber inputs drive PC simple spike firing during behavior, plastic reweighting of parallel fiber synaptic strength is a mechanism for the acquisition of cerebellar-dependent motor learning[4–6]. A $Ca^{2+}$ threshold rule governing the polarity of induced parallel fiber-to-PC synaptic plasticity[7] could provide a biochemical substrate for driving bidirectional adaptive changes to movement. However, researchers have not yet fully established a causal link between $Ca^{2+}$ elevation in PC dendrites and plasticity-induced behavioral change which precludes an understanding of the signaling mechanisms that instruct cerebellar learning.

$Ca^{2+}$ signals in PC dendrites are determined by the integrated activity of several extrinsic input sources including parallel fibers that locally evoke $Ca^{2+}$ elevation. In addition, climbing fibers from the inferior olive reliably initiate widespread $Ca^{2+}$ action potentials due to their powerful ability to depolarize PC dendrites[8]. The PC $Ca^{2+}$ response to climbing-fiber-mediated excitation is amplified when elicited in conjunction with parallel fiber activity[9] and is negatively modulated by inhibition from molecular layer interneurons[10–12]. Thus, $Ca^{2+}$ signals resulting from the activity of these diverse synaptic inputs provide PC dendrites with a repertoire of different sized transients that could differentially engage downstream enzymatic pathways for inducing opposite forms of plasticity that underlie bidirectional changes to motor output during learning[13–16]. Separate from dendritic $Ca^{2+}$ elevation, modification of PC simple spiking has also been proposed as a mechanism to instruct learning[17–19]. PC simple spike output could provide a complementary mechanism to ensure that the appropriate direction and timing of the adaptive movement is achieved for corrective behavior. Determining the contribution of these various instructive signaling mechanisms to motor learning in awake animals is imperative to resolve debates regarding the manner in which various forms of plasticity are induced in PCs under different behavioral contexts and their adaptive effect on motor output[20–24].

In this report, we optogenetically activated PCs in vivo to evoke dendritic $Ca^{2+}$ signals independent of extrinsic input sources and found that these signals were sufficient to instruct changes to coincidently active eye movements. These oculomotor responses were either increased or decreased depending on the level of induced PC $Ca^{2+}$ activity, whereas optogenetically evoked simple spiking was insufficient to impart behavioral change. Measurements in acute slices showed that optogenetically induced PC $Ca^{2+}$ transients effectively triggered plasticity induction at co-active parallel fiber synapses where the polarity of change shifts with the magnitude of $Ca^{2+}$ elevation, likely explaining the differential effect of PC $Ca^{2+}$ signaling on instructing directional changes to motor output. Optogenetically evoked plasticity induction and oculomotor adaptation were endocannabinoid receptor sensitive, indicating that this pathway acts downstream of PC dendritic $Ca^{2+}$ signaling. Together, our results establish a causal relationship between PC-autonomous signals for instructing plasticity induction and their adaptive effects on motor behavior.

## Results

### Optogenetic PC activation evokes dendritic $Ca^{2+}$ elevation.
We used optogenetic excitation of PCs to determine how their activity

instructs plasticity and motor learning. TdTomato-tagged channelrhodopsin-2 (ChR2) was conditionally expressed in PCs by crossing transgenic Ai27 mice with the *Pcp2::Cre* driver line[25,26]. In these animals, ChR2 was distributed in PCs, including a dense localization in their dendrites as evidenced by stronger reporter fluorescence in the molecular layer as opposed to the soma which appeared dim in comparison (Fig. 1a, b). PC dendrites also richly express voltage-gated $Ca^{2+}$ channels that flux $Ca^{2+}$ and mediate regenerative dendritic spiking[27–29]. Therefore, we reasoned that if cell-wide optogenetic PC activation sufficiently depolarizes the dendrite for $Ca^{2+}$-channel opening, this excitation would elevate the intracellular $Ca^{2+}$ concentration and, at a threshold level, induce dendritic spike firing that would further increase the evoked $Ca^{2+}$ response.

To explore whether direct optogenetic activation of PCs would elicit $Ca^{2+}$ signals in their dendrites, we acquired whole-cell recordings from ChR2-expressing PCs in acute cerebellar slices with a $Ca^{2+}$ indicator dye included in the patch pipette. In this manner, we were able to apply two-photon laser-scanning microscopy to measure dendritic $Ca^{2+}$ transients in response to short pulses of blue light delivered by wide-field epi-illumination (Fig. 1c). Relatively low-light powers ($<0.2\,mW/mm^2$) evoked small elevations in dendritic $Ca^{2+}$ (Fig. 1d). The magnitude of the evoked $Ca^{2+}$ transient increased with higher light powers (e.g. $>0.5\,mW/mm^2$), producing a response similar to that elicited by climbing-fiber stimulation (Fig. 1d, e). In addition, the $Ca^{2+}$ transients evoked by light-pulse trains were larger when evoked by high-light powers compared to low-light powers (Fig. 1e), indicating a consistent effect of optogenetic stimulus strength on the size of the induced dendritic $Ca^{2+}$ signals.

Although ChR2 is slightly permeable for $Ca^{2+}$[30,31], the boost in $Ca^{2+}$ transient magnitude for high-light powers points to an effect secondary to ChR2 gating. Inhibiting voltage-gated $Na^+$ channels with tetrodotoxin (TTX) did not reduce the optogenetically evoked dendritic $Ca^{2+}$ transients (Supplementary Fig. 1a, b). In fact, response amplitudes were slightly increased, a paradoxical result reflecting enhanced dendritic spiking likely owing to an indirect effect on intracellular $Ca^{2+}$ levels and SK2 channels[32]. Notwithstanding, the lack of $Ca^{2+}$ response reduction in TTX indicates that simple spiking elicited in the soma by high-power-light stimuli did not contribute to the enhanced dendritic $Ca^{2+}$ response, which is an expected result because simple spikes do not back-propagate into PC dendrites[33]. However, the evoked dendritic $Ca^{2+}$ response was greatly reduced by the addition of the P/Q-type $Ca^{2+}$ channel blocker ω-agatoxin IVA (Supplementary Fig. 1a, b). In separate experiments performed without TTX, ω-agatoxin IVA also blocked the evoked $Ca^{2+}$ transient with the addition of mibefradil further reducing the remaining response indicating that T-type $Ca^{2+}$ channels were also opened by the optogenetically induced depolarization (Supplementary Fig. 1c). This result shows that optogenetically induced excitation of PCs with high-light powers can engage the same dendritic voltage-gated $Ca^{2+}$ conductances that mediate dendritic spiking[34,35].

To directly determine whether optogenetic excitation initiates dendritic $Ca^{2+}$ action potential firing, we made multisite whole-cell electrophysiological recordings from ChR2-expressing PCs (Fig. 1f). At low powers ($<0.2\,mW/mm^2$), light stimuli depolarized both the soma and dendrite (Fig. 1g). Increasing light power led to the recruitment of compartment-specific regenerative activity including simple spiking in the soma and, with a slightly increased threshold, spike firing in the dendrite (Fig. 1g, h). By comparing results between experiments, we found that light powers suprathreshold for reliable dendritic spike initiation elicited larger $Ca^{2+}$ responses than those below the dendritic spike threshold (Fig. 1e, h). Thus, optogenetically induced $Ca^{2+}$

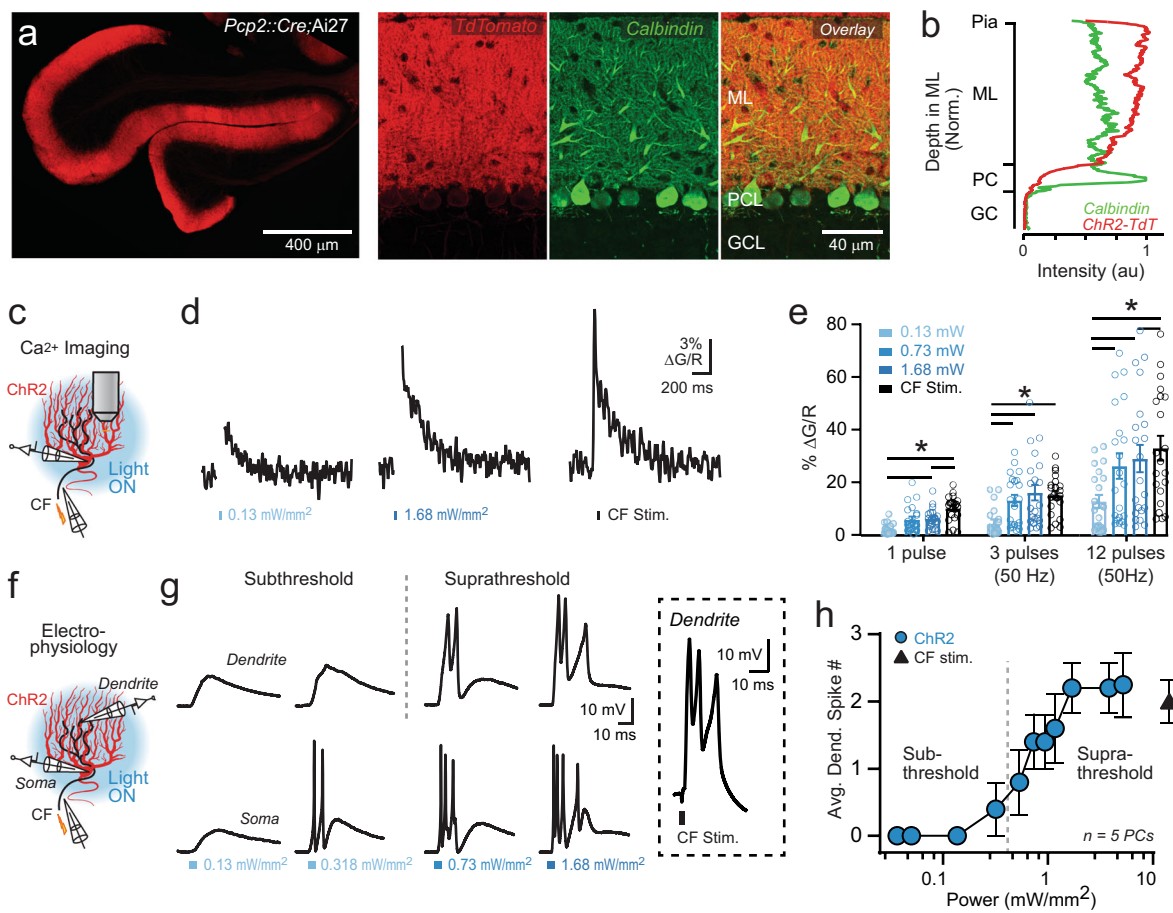

**Fig. 1 Optogenetically evoked Ca²⁺ signaling in PC dendrites. a** Representative images from a single *Pcp2::Cre;*Ai27 mouse showing tdTomato-tagged ChR2 expression in the flocculus. In the magnified view, PCs are marked by calbindin immunostaining (molecular layer, ML; Purkinje cell layer, PCL; and granule cell layer, GCL). **b** The average fluorescence intensity profile of ChR2-tdTomato in the cerebellum of an example mouse. **c** ChR2-expressing PCs were filled with Fluo-5F during whole-cell recording. Two-photon imaging was used to measure Ca²⁺ transients evoked by optogenetic excitation or climbing-fiber (CF) stimulation. **d** Average Ca²⁺ transients from the same PC dendrite evoked by single pulses of light ($\lambda$ 461 nm; 5 ms; artifacts blanked for clarity), shown relative to the climbing-fiber-evoked response. **e** Summary plot of peak Ca²⁺ transient amplitude for different stimulus conditions. In addition to single stimuli, trials also included bursts of closely spaced light pulses to elicit activity ($n = 21$ dendritic sites obtained from seven PCs; six mice). Descriptive statistics: 1 pulse: 0.13 vs. 1.68 mW $P = 0.0345$, 0.13 mW vs. CF, $P < 0.0001$, 0.73 mW vs. CF, $P = 0.0404$; 3 pulses: 0.13 vs. 0.73 mW, $P < 0.0001$, 0.13 vs. 1.68 mW, $P < 0.0001$, 0.13 mW vs. CF, $P < 0.0001$; 12 pulses: 0.13 vs. 0.73 mW, $P < 0.0001$, 0.13 vs. 1.68 mW, $P < 0.0001$, 0.13 mW vs. CF, $P < 0.0001$, 0.73 vs. CF, $P = 0.0002$). **f** Simultaneous patch-clamp recordings from the soma and dendrite of ChR2-expressing PCs allowed for high-resolution measurement of optogenetically induced electrogenic activity. **g** Electrophysiological responses from the same PC to single pulses of light at increasing powers ($\lambda$ 461 nm; 5 ms). The evoked dendritic response to climbing-fiber stimulation is shown for a separate cell. **h** Plot shows the average number of optogenetically evoked dendritic spikes as a function of light power; the threshold for evoking reliable dendritic spiking is indicated by the dashed line. All data are mean ± SEM; asterisk indicates $P < 0.05$, two-way repeated-measures ANOVA with Tukey's post test.

signaling in PC dendrites is amplified by regenerative electrogenic activity. Notably, at high-light powers (>1.0 mW/mm²), the optogenetically evoked electrical waveform in the dendrite closely resembled the response elicited by climbing-fiber-mediated excitation (Fig. 1g, h), an input that drives somatically uncoupled bursts of dendritic Ca²⁺ action potentials[36]. Together, these results indicate that optogenetic PC activation produces a range of dendritic Ca²⁺ signals, which are partially dependent on amplification by regenerative dendritic spiking; an electrogenic response mimicking those elicited by climbing-fiber-evoked excitation.

**Optogenetic PC activation induces synaptic plasticity.** Ca²⁺ elevation in PC dendrites is a biochemical trigger for plasticity induction. These signals drive lasting changes in the efficacy of recently active parallel fiber synapses, with the different polarities

of alteration having varying Ca²⁺ thresholds of induction. Parallel fiber-to-PC long-term depression (LTD) requires larger increases in dendritic Ca²⁺ levels relative to long-term potentiation (LTP)[7]. Hence, LTD is preferentially induced by the amplified dendritic Ca²⁺ response to climbing-fiber-mediated excitation[9,37]. To characterize how ChR2-induced Ca²⁺ signaling in PC dendrites instructs plasticity induction, we compared the amplitude of test parallel fiber-evoked excitatory postsynaptic potentials (EPSPs) before and after repeated tetanization in conjunction with optogenetic stimuli. LTP was induced after the parallel fiber tetanus was paired with an optogenetic stimulus subthreshold for dendritic spiking (1.32 ± 0.09 of baseline; $P = 0.0005$; Wilcoxon test; Fig. 2a, b). Plasticity induction was dependent on optogenetic-induced activity because pairing the parallel fiber tetanus with light alone in non-ChR2-expressing control PCs failed to induce a change in parallel fiber synaptic strength (0.99 ± 0.11 of baseline; $P = 0.94$; Wilcoxon test; Fig. 2b).

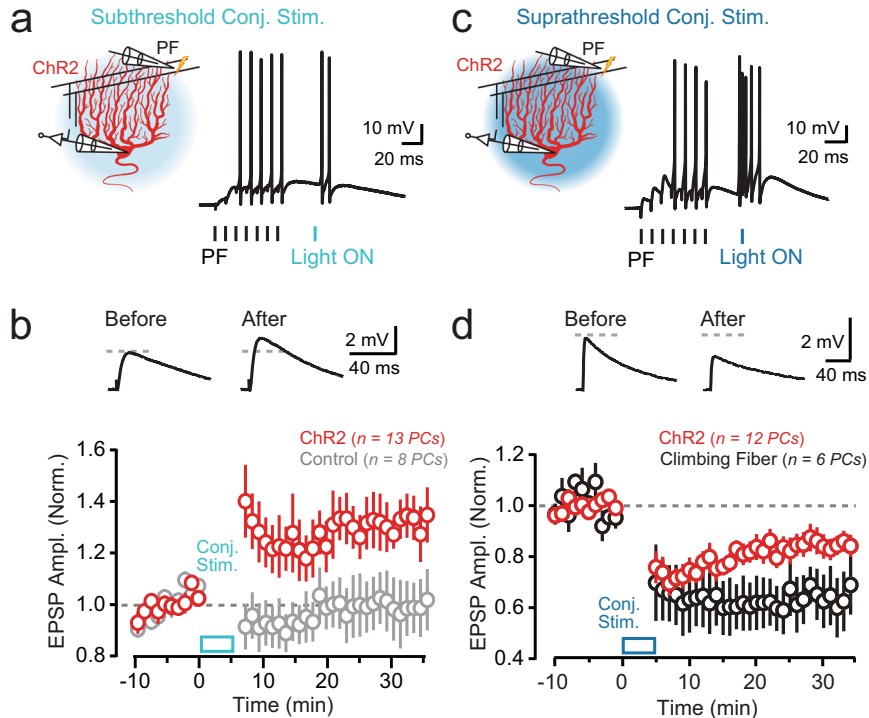

**Fig. 2 Optogenetic PC activation induces synaptic plasticity. a** During PC recordings, a parallel fiber (PF) tetanus (100 Hz, 70 ms) preceded an optogenetically induced stimulus ($\Delta$ 25 ms); light power was subthreshold for evoking dendritic spike firing ($\lambda$ 461 nm; 5 ms; 0.15 mW/mm$^2$). This pairing was delivered for 300 repetitions (1 Hz). **b** Top: average test EPSPs to parallel fiber stimulation before and after the conjunctive pairing procedure. Bottom: plot of change in EPSP amplitude across PCs shows optogenetically induced LTP (red). Test EPSP amplitude was not affected by the light-pairing procedure when recording in non-ChR2-expressing control PCs (gray). Data were obtained from 12 mice. **c** In separate recordings, the parallel fiber tetanus was paired with an optogenetic stimulus suprathreshold for evoking dendritic spike initiation ($\lambda$ 461 nm; 5 ms; 1.7 mW/mm$^2$). **d** Top: average test EPSPs before and after the conjunctive pairing procedure with high-power light. Bottom: summary plot of EPSP amplitude across PCs shows optogenetically induced LTD (red), the same polarity of plasticity induced by pairing the parallel fiber tetanus with a climbing-fiber stimulus in place of the light pulse (black). Data were obtained from 14 mice. In the plots, data are presented as mean ± SEM.

We also found that optogenetic PC activation suprathreshold for dendritic spiking effectively induced plasticity. However, when paired with parallel fiber tetanus, this stimulus yielded LTD instead of LTP ($0.84 \pm 0.04$ of baseline; $P = 0.0049$; Wilcoxon test; Fig. 2c, d). Similarly, LTD was induced when the same parallel fiber tetanus was paired with climbing-fiber stimulation in place of a light pulse ($0.64 \pm 0.10$ of baseline; $P = 0.03$; Wilcoxon test; Fig. 2d). However, the change in synaptic efficacy obtained by conjunctive pairing with climbing fibers was greater than that obtained by optogenetically induced PC activation ($P = 0.04$, $t$ test). In support of a Ca$^{2+}$ threshold rule governing plasticity induction at parallel fiber-to-PC synapses, these results indicate that amplified Ca$^{2+}$ signaling through dendritic spiking, whether induced by high-intensity optogenetic stimuli or climbing-fiber-induced excitation, drives LTD whereas more moderate Ca$^{2+}$ signals result in LTP.

**Optogenetic PC activation in awake mice.** Cerebellar-dependent motor learning depends on signaling that instructs appropriate forms of plasticity to accurately recalibrate movements in response to motor errors. Corrective behavior is achieved by strengthening or weakening the motor output, depending on the context of the error. Although opposite-polarity changes in the efficacy of parallel fiber-to-PC synapses signaled by a Ca$^{2+}$ threshold rule provide a mechanism for imparting bidirectional alterations in motor output, the causal effect of such Ca$^{2+}$ signals on behavior remains unclear. Therefore, we used optogenetic activation of PCs to determine how different levels of dendritic

Ca$^{2+}$ signaling during movements translate to plasticity-induced changes in motor performance.

In our approach, we examined the vestibulo-ocular reflex (VOR), a nonvolitional eye moment elicited in response to head motion. A well-calibrated VOR helps maintain a stable gaze. Hence, when visual motion (retinal slip) is encountered during a vestibular stimulus, the mismatch is interpreted as a motor error and triggers the induction of corrective plasticity in the cerebellar flocculus to restore distortion-free performance[38]. Retinal slip can elicit adaptive increases or decreases in VOR amplitude (gain); in this process, the direction of learned change encoded in the cerebellum depends on whether visual motion occurs in the opposite or same direction of the head motion[39]. However, by training mice in darkness, we eliminated visual feedback. Instead of using retinal slip to drive instructional signaling, we stimulated ChR2-expressing PCs with light pulses through bilateral optical fiber implants targeting both flocculi in conjunction with head turns when parallel fibers fire during vestibular motion (Fig. 3a). This approach allowed us to determine whether the pairing of this activity results in changes to VOR performance. Importantly, to account for VOR gain changes due to darkness-induced habituation[40], we quantified the adaptive effect of optogenetic PC activation on VOR performance by comparing induced changes during training to those for sessions consisting of vestibular motion alone[18].

To first establish whether optogenetically induced PC excitation can effectively elicit Ca$^{2+}$ activity in vivo, we transduced floccular PCs of *Pcp2::Cre* mice with adeno-associated viruses

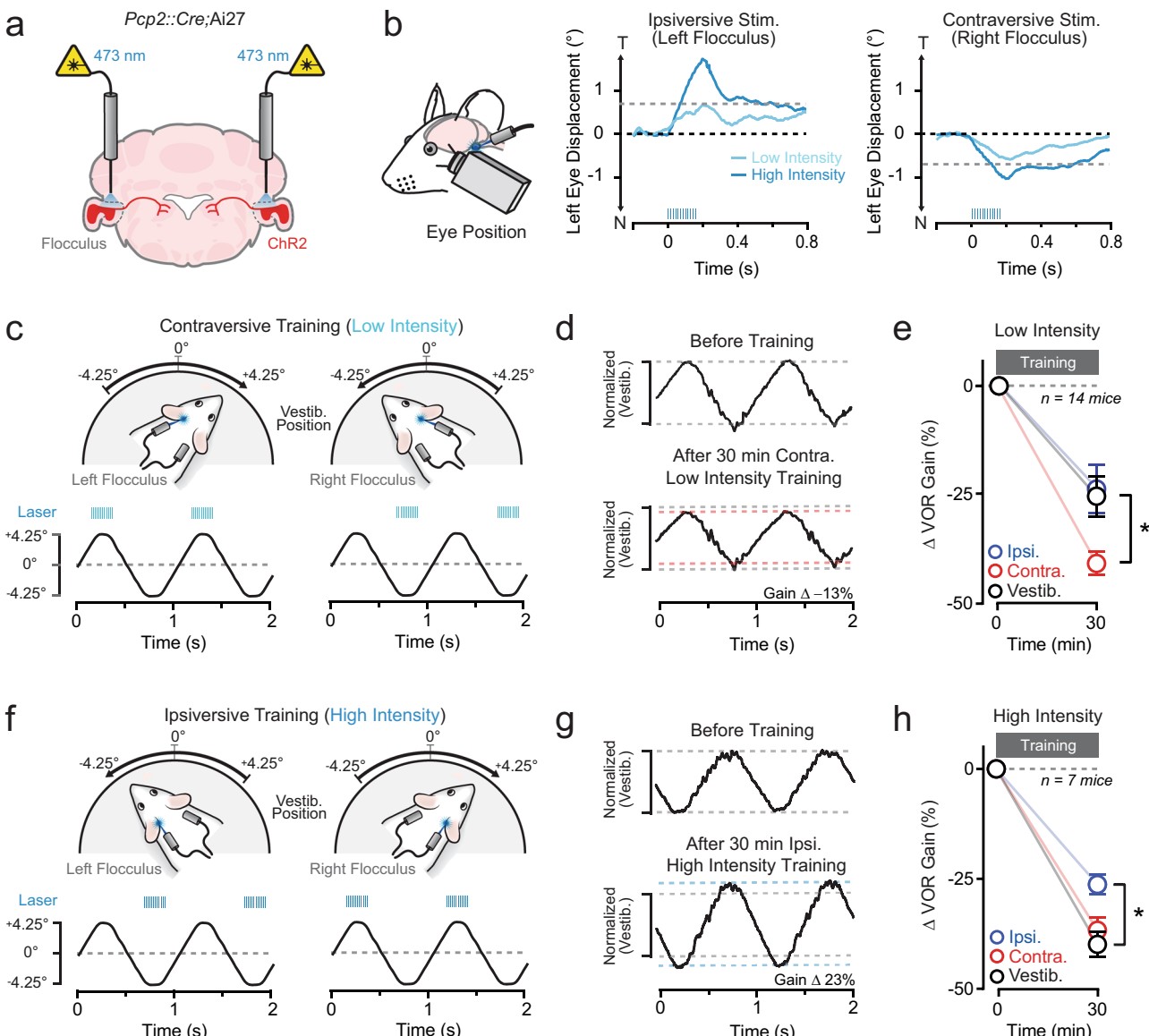

**Fig. 3 Optogenetic PC activation induces VOR learning. a** Optical fibers targeting both flocculi were bilaterally implanted into *Pcp2::Cre;*Ai27 mice to stimulate ChR2-expressing PCs. **b** Eye movements in a mouse evoked by unilateral optogenetic PC activation using different light powers ($λ$ 471 nm; 12 pulses; 5 ms; 50 Hz, 1 and 2 mW). The direction of lateral eye displacement (T, temporal; N, nasal) depended on whether PCs in the ipsiversive or contraversive flocculus were stimulated (laser pulses indicated by blue tick marks). The absolute amplitude of evoked movements was used to categorize the intensity of optogenetic stimuli (> or <0.75°, high- and low-intensity, respectively). **c** In darkness, head-fixed mice were trained by pairing sinusoidal vestibular motion (1 Hz) with a low-intensity optogenetic stimulus ($λ$ 471 nm; 12 pulses; 5 ms; 50 Hz) beginning near the completion of head turns (training lasted 30 min). In the example traces below, PC activation was timed to the end of contraversive vestibular motion (average head position, black trace; laser pulses; blue tics). **d** Average VOR-evoked eye movements measured before and after the mouse were trained with a pairing procedure using low-intensity optogenetic stimuli timed to the end of contraversive vestibular motion. The traces were normalized to that of time-matched, control responses recorded during sessions of training consisting of the vestibular stimulus alone. **e** The effect of pairing either contraversive or ipsiversive vestibular motion with low-intensity optogenetic stimuli on VOR gain, shown relative to control sessions. Measurements were obtained from the same mice. Changes are shown normalized to a baseline measurement obtained immediately before training ($t = 0$ min) for each condition. Vestibular stimulation vs. ipsiversive stimulation: $P = 0.6903$; vestibular stimulation vs. contraversive stimulation: $P = 0.0388$. **f** In separate sessions, a cohort of the same mice was trained using pairing that included high-intensity optogenetic stimuli ($λ$ 471 nm; 12 pulses; 5 ms; 50 Hz). In the example head-position traces below, PC activation was timed to the end of ipsiversive vestibular motion. **g** Average VOR eye movements before and after training with high-intensity optogenetic stimuli timed to the end of ipsiversive vestibular motion. **h** The effect of pairing vestibular motion with high-intensity optogenetic PC stimulation on VOR gain across mice. Vestibular stimulation vs. ipsiversive stimulation: $p = 0.0068$, Vestibular stimulation vs. contraversive stimulation: $P = 0.1151$. Data are shown mean ± SEM with an asterisk indicating $P < 0.05$, two-way repeated-measures ANOVA with Dunnett's post tests.

(AAVs) containing Cre-dependent ChR2 and jRCaMP1a, a red $Ca^{2+}$ sensor[41], and measured blue light-evoked $Ca^{2+}$ responses using photometry through an implanted optical fiber (Supplementary Fig. 2a, b). In head-fixed quiescent mice, brief bursts of light pulses elicited $Ca^{2+}$ transients. The magnitude of the evoked response increased with higher light powers (Supplementary Fig. 2c, d), indicating that the level of PC excitation was sensitive to the optogenetic stimulus strength. Optogenetic PC activation also elicited eye movements time-locked to the light stimulus; movement amplitude exhibited a similar increase with light power (Fig. 3b and Supplementary Fig. 3a–c). Optogenetically elicited motor responses are indicative of evoked PC activity due to increased simple spike firing which inhibits target neurons in the vestibular nuclei controlling eye movement[17,18,42]. Therefore, to normalize the level of optogenetically induced PC $Ca^{2+}$ signaling between training conditions and to facilitate comparisons across animals, we calibrated the intensity of optogenetic stimuli using the absolute amplitude of evoked eye movements during quiescence for each animal (Fig. 3b).

**Optogenetic PC activation induces motor learning**. Pairing vestibular motion (1 Hz sinusoids) in conjunction with a low-intensity optogenetic stimulus was effective at instructing a learned change in VOR performance when compared to control sessions of vestibular motion alone. However, the ability to produce this change was specific to the timing of the optogenetic stimulus relative to the direction of the head turn. PC activation at the end of contraversive vestibular motion (i.e., when PCs in the left flocculus were stimulated after rightward head turns and PCs in the right flocculus were stimulated after leftward head turns) resulted in a learned decrease in VOR gain (Fig. 3c–e). However, optogenetically induced PC excitation at the end of ipsiversive vestibular motion (i.e., when PCs in the left flocculus were stimulated after leftward head turns and PCs in the right flocculus were stimulated after rightward head turns) had no discernable effect on VOR gain (Fig. 3e).

To determine whether VOR performance changes resulting from optogenetically evoked PC excitation are sensitive to the level of induced activity, we trained a cohort of the same mice in separate sessions by pairing vestibular motion with a high-intensity optogenetic stimulus (Fig. 3b). In this condition, PC activation at the end of ipsiversive vestibular motion was effective for instructing learning. However, this pairing drove an increase rather than a decrease in VOR gain, in comparison to control training sessions of vestibular motion alone (Fig. 3f–h). Surprisingly, high-intensity optogenetic PC activation timed to the end of contraversive vestibular motion was not effective for inducing learned changes to VOR gain (Fig. 3h). Comparisons across mice showed a significant difference in induced changes to VOR performance for low- and high-intensity optogenetic stimuli timed to either the ipsiversive or contraversive phase of vestibular motion (Supplementary Fig. 4a, b, respectively). Together, these results establish that PC-autonomous activity that drives dendritic $Ca^{2+}$ elevation can instruct bidirectional changes to movement strength, dependent on the level of induced excitation. However, the context of vestibular motion influences the effectiveness of these signals in imparting behavioral change, indicating a conditional sensitivity of cerebellar circuitry to encoding different types of learning.

Under natural behavior, learned increases in VOR gain arise when retinal slip occurs in the direction opposite to that of head motion. This context modulates climbing-fiber firing during ipsiversive vestibular motion[43]. This finding suggests that high-intensity optogenetic PC activation may substitute for dendritic $Ca^{2+}$ signals induced by climbing-fiber-mediated excitation. In accordance with this idea, pairing head turns with optogenetic stimulation of ChR2-expressing climbing fibers, transduced by an AAV injected into the inferior olive of wildtype mice (Supplementary Fig. 5a), resulted in a learned increase in VOR gain when light pulses were timed to the end of ipsiversive vestibular motion, compared to control training sessions (Supplementary Fig. 5b–d). Thus, high-intensity optogenetic PC activation and climbing-fiber-evoked excitation likely engage convergent $Ca^{2+}$-dependent plasticity mechanisms to impart similar alterations to VOR performance.

**Optogenetically induced plasticity and learning depend on endocannabinoid signaling**. In response to dendritic $Ca^{2+}$ elevation, PCs release endocannabinoids that bind to $CB_1$ receptors expressed on presynaptic parallel fiber boutons[44]. As both LTP and LTD induction at parallel fiber-to-PC synapses depend on $CB_1$ receptor activation[45,46], we reasoned that retrograde endocannabinoid signaling is a potential mechanism converting PC dendritic $Ca^{2+}$ activity into plasticity that imposes behavioral change. To test this idea, we first confirmed that optogenetically induced alteration of parallel fiber synaptic strength is endocannabinoid mediated by repeating ex vivo plasticity experiments in the presence of the $CB_1$ receptor antagonist AM251. With $CB_1$ receptors blocked, pairing parallel fiber tetanus with light stimulation failed to induce a change in synaptic efficacy; AM251 inhibited plasticity regardless of whether the light stimulus was subthreshold or suprathreshold for dendritic spiking ($1.07 \pm 0.13$ and $0.97 \pm 0.07$ of baseline; low- and high-power-light pairing in AM251, respectively; $P = 0.46$ and $P = 0.99$; Wilcoxon tests; Fig. 4a–c). This result demonstrates a common requirement for endocannabinoid signaling in instantiating distinct types of parallel fiber plasticity resulting from different levels of optogenetically evoked $Ca^{2+}$ elevation in PC dendrites.

To probe for a link between dendritic $Ca^{2+}$ signaling and forms of plasticity induction that mediate bidirectional VOR learning, we administered AM251 to ChR2-expressing mice by intraperitoneal injection to block $CB_1$ receptors in vivo[47]. We then trained these animals by pairing vestibular stimuli with optogenetically induced activation of floccular PCs in the contexts that we previously observed induced learning (Fig. 4d). Unlike control training sessions in which these same animals were administered saline and the solvent DMSO, neither low-intensity optogenetic stimuli at the end of contraversive head turns nor high-intensity optogenetic stimuli at the end of ipsiversive head turns were effective in instructing learned decreases or increases in VOR amplitude, respectively, relative to that elicited in sessions of vestibular motion-only training in the same pharmacological condition (in AM251: contraversive low-intensity optogenetic stimulation vs vestibular motion alone only; $P = 0.62$ ipsi. high-intensity optogenetic stimulation vs vestib. only; $P = 0.42$; paired $t$ tests; Fig. 4e, f). As $CB_1$-receptor sensitivity is observed for both LTP and LTD induction and for optogenetically induced bidirectional changes to VOR performance, our results indicate that dendritic $Ca^{2+}$ elevation engages endocannabinoid signaling, resulting in opposite-polarity plasticity at parallel fiber inputs that were active during the vestibular stimulus.

**Optogenetically induced plasticity requires dendritic $Ca^{2+}$ signaling**. In addition to evoking dendritic $Ca^{2+}$ transients, optogenetic PC excitation also drives somatic simple spike firing. This combination of induced activity presents a potential confound to the interpretation of our behavioral results because the modulation of PC simple spiking is a candidate instructive signal for instantiating motor learning[17–19,48]. Therefore, to disambiguate the potential effects of optogenetically evoked PC

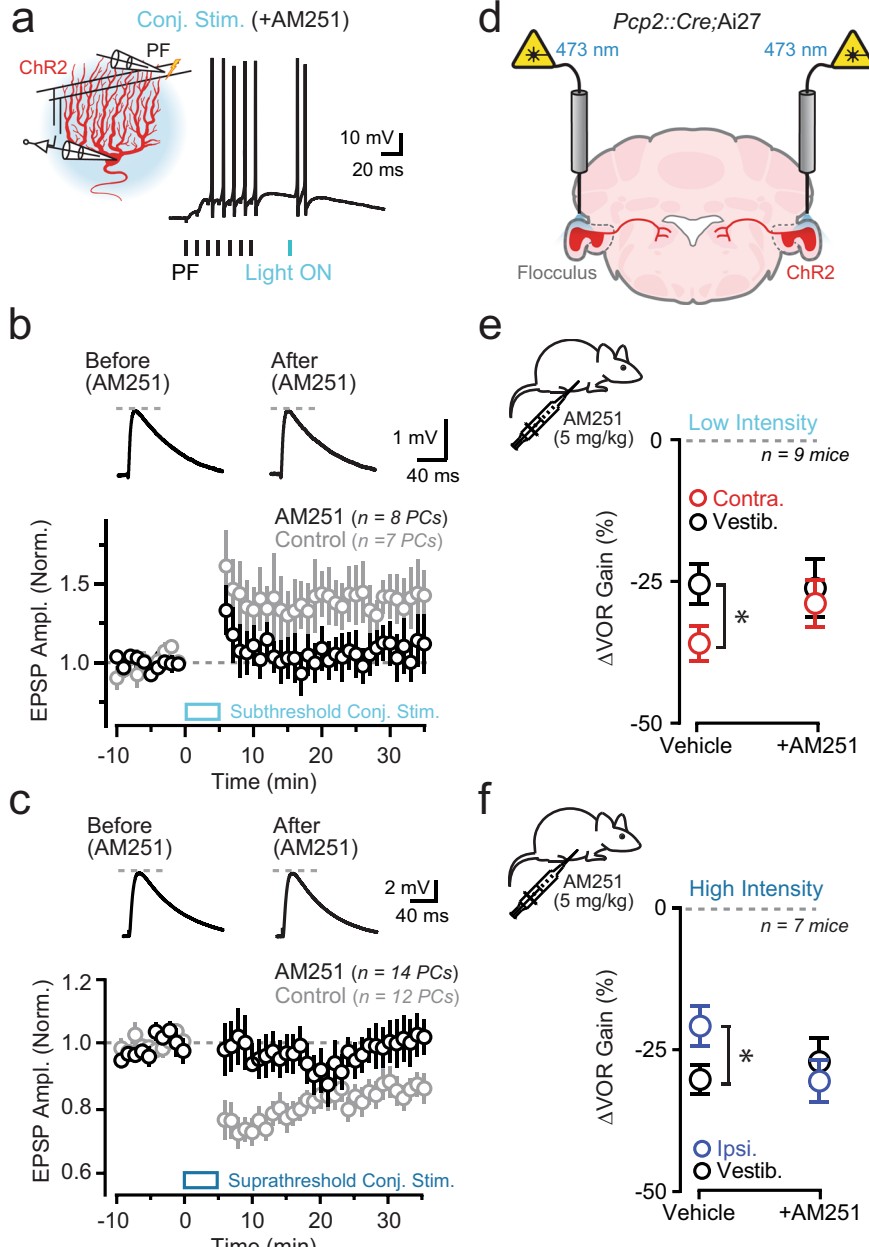

**Fig. 4 Optogenetically induced plasticity and learning requires endocannabinoid signaling. a** In the presence of the CB$_1$ receptor antagonist AM251 (5 μM), a parallel fiber (PF) tetanus (100 Hz, 70 ms) was repeatedly paired (300 repetitions; 1 Hz) with a light pulse (λ 461 nm; 5 ms; either 0.13 or 1.7 mW/mm$^2$) to activate ChR2-expressing PCs. **b, c** Top: average parallel fiber-evoked EPSPs recorded in PCs before and after conjunctive pairing with an optogenetic stimulus either subthreshold or suprathreshold for dendritic spike firing. Bottom: plots of average EPSP amplitude across PCs for control recordings (gray) or in the presence of AM251 (black). Data obtained from 24 mice total. **d** During training, ChR2-expressing PCs in the flocculus were activated using optogenetic stimuli (12 pulses, 5 ms, 50 Hz) in conjunction with vestibular motion (1 Hz). Either vehicle (containing saline + DMSO) or AM251 (5 mg/kg in saline + DMSO) was administered 20 min prior to training. **e, f** Summary data for mice trained with either low- or high-intensity optogenetic stimuli timed to the end of contraversive or ipsiversive vestibular motion, respectively, after being administered vehicle or AM251. Separate sessions also included vestibular-only training in the same pharmacological conditions. Low-intensity stimulation, vehicle: vestibular vs. contraversive stimulation, $P < 0.05$; AM251: vestibular stimulation vs. contraversive stimulation, $P > 0.05$. High-intensity stimulation, vehicle: vestibular vs. ipsiversive stimulation, $P < 0.05$; AM251: vestibular vs. ipsiversive stimulation, $P > 0.05$. All data are mean ± SEM; asterisk indicates $P < 0.05$, two-way repeated measures with Sidak's post test.

simple spiking on induced changes to VOR performance from those attributable to the dendritic Ca$^{2+}$ response, we avoided expressing ChR2 in PC dendrites that drive local Ca$^{2+}$ elevation in their arbors. We achieved this by attaching the somatic targeting motif of the K$_v$2.1 K$^+$ channel to the *c*-terminus of ChR2 (ChR2-K$_v$2.1)[49–51].

AAV-mediated transduction of Cre-dependent ChR2-K$_v$2.1 in *Pcp2::Cre* mice resulted in a predominant somatic localization of ChR2 in PCs, although there was a modicum of expression in their proximal dendrites (Fig. 5a). As shown in cerebellar slice recordings from PCs expressing ChR2-K$_v$2.1, short light pulses drove well-timed somatic simple spikes without evoking

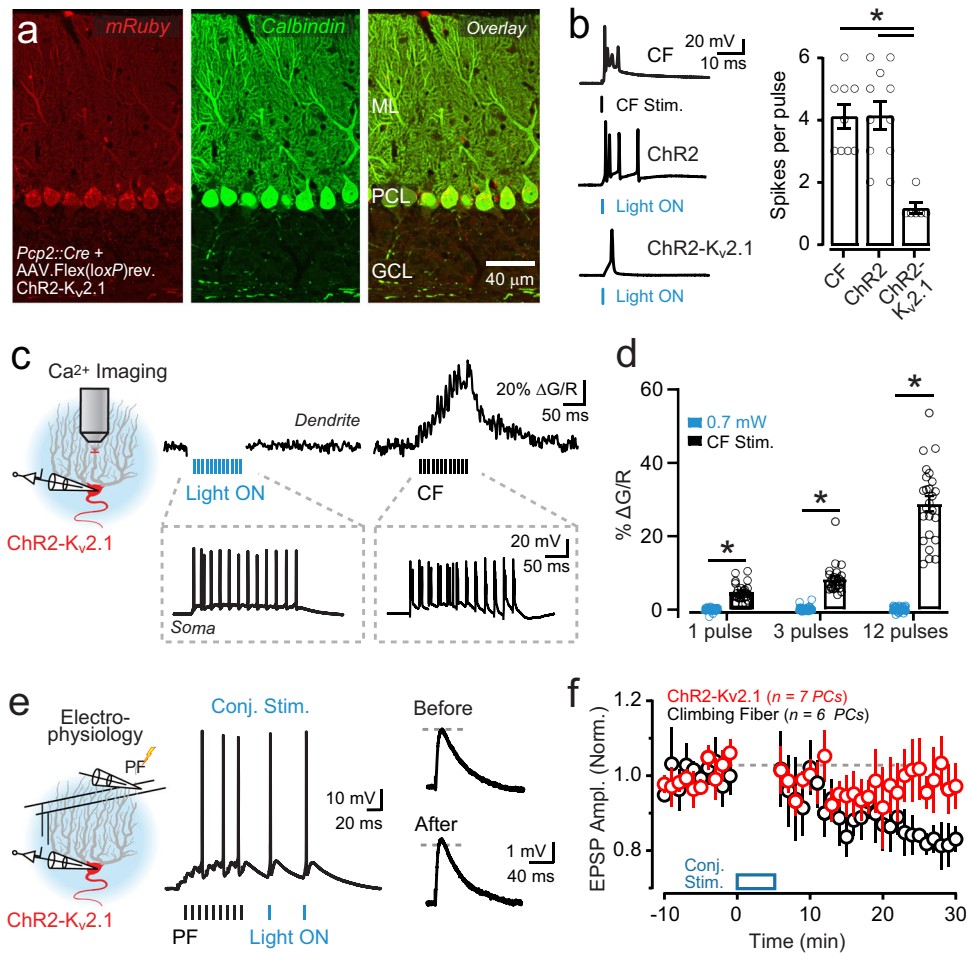

**Fig. 5 Absence of optogenetically induced dendritic Ca$^{2+}$ signaling and plasticity in PCs with soma-restricted ChR2 expression. a** Image from a *Pcp2::Cre* mouse expressing mRuby-tagged ChR2-K$_v$2.1 by Cre-dependent AAV transduction; PCs are marked by calbindin immunostaining. **b** Electrophysiological recordings from the soma of two different PCs, expressing either ChR2 ($n = 11$ PCs) or ChR2-K$_v$2.1 ($n = 6$ PCs), in response to single pulses of light (5 ms; 0.7 mW/mm$^2$). A complex spike in the ChR2-expressing cell was evoked by climbing-fiber (CF) stimulation for comparison ($n = 9$ PCs). The summary plot on the right shows the average somatic spiking response to the different stimuli (one-way ANOVA with Tukey's post test; CF vs. ChR2: $P > 0.05$, CF vs. ChR2-Kv2.1: $P \leq 0.001$, ChR2 vs. ChR2-Kv2.1: $P \leq 0.001$). Data from 14 mice. **c** Dendritic Ca$^{2+}$ activity in a ChR2-K$_v$2.1-expressing PC to a burst of optogenetic stimuli (12 pulses; 5 ms; 50 Hz; 0.7 mW/mm$^2$) or to repeat climbing-fiber activation. Somatic spiking is shown in the gray boxes below. **d** Lack of dendritic Ca$^{2+}$ signals in response to optogenetic stimuli in PCs expressing ChR2-K$_v$2.1 ($n = 25$ dendrites, four cells, three mice; two-way repeated-measures ANOVA with Dunnett's post test; 1 pulse: CF vs. ChR2-Kv2.1, $P < 0.0001$, 3 pulses: CF vs. ChR2-Kv2.1, $P < 0.0001$, 12 pulses: CF vs. ChR2-Kv2.1, $P < 0.0001$). **e** A parallel fiber (PF) tetanus (100 Hz; 70 ms) was stimulated in conjunction with optogenetic activation of ChR2-K$_v$2.1-expressing PCs (2 pulses; 5 ms; 20 Hz); pairing was repeated 300 times (1 Hz). On the right, average parallel fiber-evoked EPSPs are shown before and after conjunctive pairing. **f** In recordings from ChR2-K$_v$2.1-expressing PCs, EPSP amplitude remained unchanged following the conjunctive parallel fiber-light-pairing procedure (red) as opposed to a depression of EPSP amplitude following the parallel fiber -CF pairing procedure (black). Data were obtained from ten mice. All data are mean ± SEM; asterisk indicates $P < 0.05$.

burst-like waveforms that mimic climbing-fiber-evoked complex spikes as observed in response to ChR2-mediated somatodendritic excitation (Fig. 5b). Importantly, although light-pulse trains drove simple spike firing in ChR2-K$_v$2.1-expressing PCs, these optogenetic stimuli resulted in a negligible increase in dendritic Ca$^{2+}$ (Fig. 5c, d). Climbing-fiber-evoked transients confirmed the dendritic Ca$^{2+}$ responsiveness of these same cells (Fig. 5c).

As parallel fiber-to-PC LTP and LTD are both Ca$^{2+}$ dependent[7,52], we expected that restricting ChR2 from PC dendrites would prevent optogenetically induced synaptic plasticity due to the absence of an evoked dendritic Ca$^{2+}$ response. In support of this hypothesis, pairing parallel fiber tetanus with optogenetic activation of PCs expressing ChR2-K$_v$2.1 failed to induce a measurable change in parallel fiber synaptic strength, even though we applied several light pulses in an attempt to increase the overall level of optogenetically induced

excitation ($0.93 \pm 0.04$ of baseline; $P = 0.25$; Wilcoxon test; Fig. 5e, f). The conjunctive pairing of parallel fiber and climbing-fiber activity continued to induce LTD in ChR2-K$_v$2.1-expressing PCs demonstrating a susceptibility for plasticity induction in these cells ($0.78 \pm 0.02$ of baseline; $P = 0.03$; Wilcoxon test; Fig. 5f). Thus, by using a subcellular targeting strategy to limit ChR2 expression to the PC soma, we preserved optogenetically induced simple spiking but avoided evoking dendritic Ca$^{2+}$ elevation, which, in turn, prevented plasticity induction at parallel fiber inputs.

**Absence of motor learning in response to optogenetically induced PC simple spiking.** To address whether optogenetically induced PC simple spiking is sufficient to instruct VOR learning independent of evoked dendritic Ca$^{2+}$ signaling, we trained

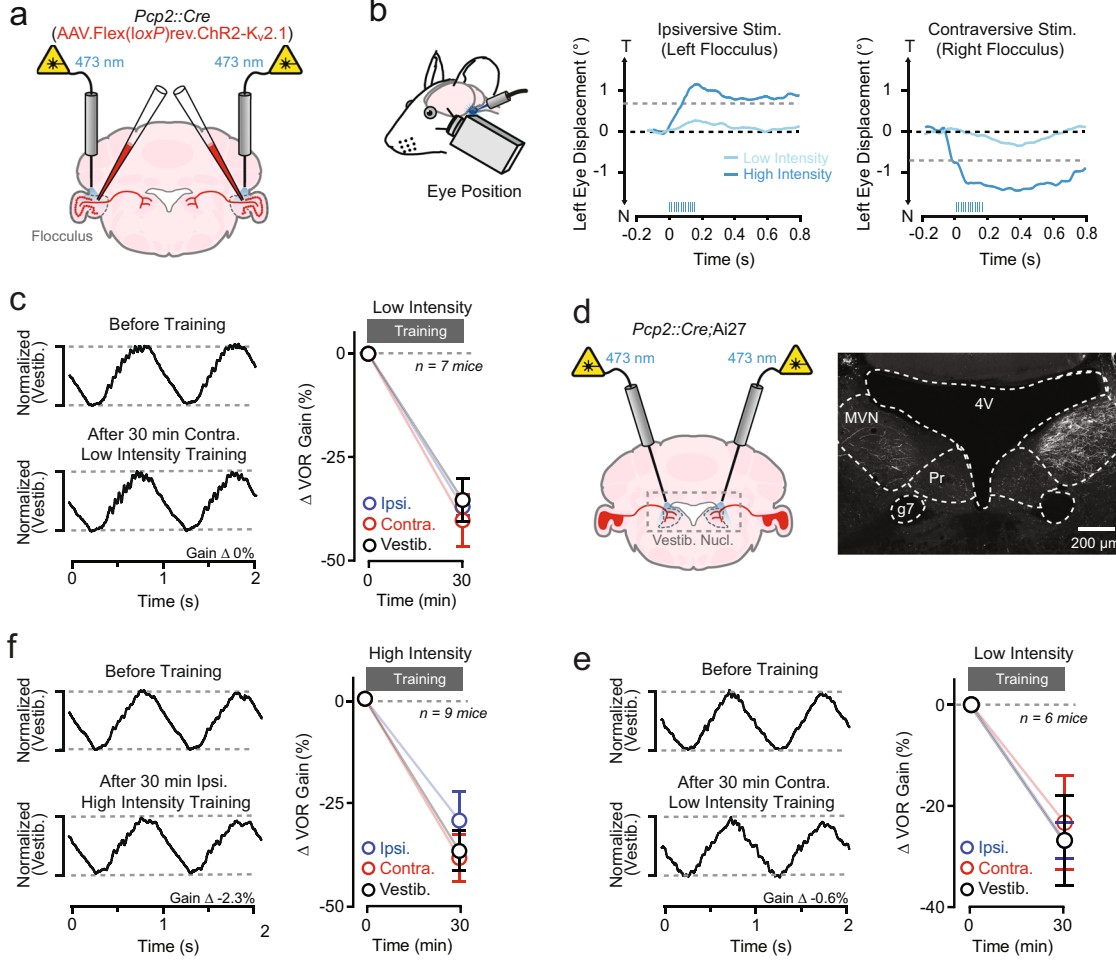

**Fig. 6 Absence of motor learning with optogenetically induced PC simple spike firing. a** *Pcp2::Cre* mice were implanted with bilateral optical fibers to stimulate floccular PCs expressing soma-targeted ChR2-Kv2.1 by AAV-mediated transduction. **b** To calibrate the intensity of the optogenetic stimulus, evoked eye movements were monitored in response to activation of ChR2-Kv2.1-expressing PCs (12 pulses, 5–10 ms; 50 Hz). **c** Average VOR-evoked eye movements measured before and after training with low-intensity optogenetic stimuli timed to the end of contraversive vestibular motion. The traces were normalized to that of time-matched, control responses recorded during training sessions consisting of vestibular motion alone. The summary plot shows the lack of effect of pairing low-intensity optogenetic PC activation with vestibular motion on VOR gain. **d** *Pcp2::Cre;Ai27* mice were implanted with bilateral optical fibers targeting the medial vestibular nuclei (MVN) to activate the ChR2-expressing axon projections of floccular PCs. The image on the right shows ChR2-tdTom expressing PC terminals in the MVN (fourth ventricle, 4 V; genu of CN VII, g7). **e** VOR-evoked eye movements before and after pairing low-intensity optogenetic activation of PC axons with vestibular motion on normalized VOR performance. Summary plot show a lack of an effect of both contraversive and ipsiversive pairing contexts compared to control training sessions. Data are presented as mean ± SEM and analyzed with two-way repeated-measures ANOVA with Dunnett's post test. **f** Comparison of VOR performance before and after pairing with high-intensity optogentic stimulation of ChR2-Kv2.1-expressing PCs in the flocculus, timed to the end of ipsiversive vestibular motion. Summary data from all mice is shown on the right.

*Pcp2::Cre* mice with ChR2-Kv2.1-expressing floccular PCs by pairing vestibular stimuli with somatic PC activation (Fig. 6a). Prior to training, we first calibrated the level of optogenetically induced activity in each mouse using the absolute amplitude of evoked eye movement elicited in quiescence (Fig. 6b). This step ensured that we stimulated ChR2-Kv2.1-expressing PCs to an extent similar to that achieved by ChR2-mediated somatodendritic excitation during training. We found that low-intensity optogenetic stimuli timed to the end of either contraversive or ipsiversive vestibular motion failed to induce a change in VOR gain, relative to control sessions with vestibular motion-only training (Fig. 6c). This result indicates that learned decreases in VOR gain do not result from optogenetically evoked simple spiking. To further evaluate this possibility, we bilaterally implanted optical fibers targeting both vestibular nuclei to activate ChR2-expressing axon terminal projections of floccular PCs in *Pcp2::Cre;Ai27* mice (Fig. 6d). By bypassing the flocculus, we

circumvented depolarizing dendrites when exciting PCs and hence avoided elevating Ca²⁺ levels in this subcellular compartment. Light pulses delivered to the vestibular nuclei evoked small eye movements, confirming that optogenetically induced axonal excitation was sufficient to drive PC simple spike firing. However, relative to control sessions of vestibular motion-only training, the amplitude of the VOR remained unchanged after training sessions that including pairing with low-intensity optogenetic PC activation at the end of either contraversive or ipsiversive vestibular stimuli (Fig. 6e). Thus, this stimulus was ineffective in instructing gain-decrease learning.

We also trained the same cohort of ChR2-Kv2.1-expressing mice using high-intensity optogenetic stimuli in the flocculus (Fig. 6f). However, somatic PC activation through floccular-targeted excitation, timed to the end of either contraversive or ipsiversive vestibular motion, also failed to affect VOR performance. Thus, the absence of learning due to PC activity paired

with vestibular motion was not dependent on a threshold level of induced PC excitation. Altogether, these results conclusively demonstrate that activity-induced PC simple spiking over the time course of training (30 min) is ineffective for instructing learned changes to VOR performance. By ruling out the possibility that VOR learning results from an induced increase in PC simple spike output, this finding further substantiates that optogenetically elicited dendritic $Ca^{2+}$ signaling and endocannabinoid-mediated plasticity induction are responsible for instructing motor learning in response to optogenetic PC activation.

## Discussion

In this study, we found that ChR2-mediated excitation of PCs produces dendritic $Ca^{2+}$ transients that are amplified by depolarization-evoked dendritic spiking. These $Ca^{2+}$ signals triggered LTP or LTD at co-active parallel fiber synapses; the polarity of plasticity induction followed a $Ca^{2+}$ threshold rule. In vivo, optogenetically evoked PC excitation was sufficient to instruct motor learning. The level of induced PC $Ca^{2+}$ activity governed the direction of movement modification, though the effectiveness of these signals to impart behavioral change depended on the context in which the $Ca^{2+}$ response was elicited. Endocannabinoid signaling, engaged downstream of the optogenetically evoked dendritic $Ca^{2+}$ response, was required for both opposite-polarity plasticity induction and bidirectional motor learning. Optogenetically evoked simple spiking in PCs, elicited over the time course of training using soma-targeted ChR2 expression, was insufficient to induce an adaptive behavioral change, pointing to the primacy of dendritic $Ca^{2+}$ signaling in rapidly instructing bidirectional cerebellar motor learning.

**PC dendritic $Ca^{2+}$ signals direct opposite-polarity forms of plasticity induction**. Our results show that PC-wide ChR2 activation effectively evokes dendritic $Ca^{2+}$ signals by depolarizing the local membrane potential in a light-dependent manner resulting in the opening of voltage-gated $Ca^{2+}$ channels. By altering the level of optogenetic-induced excitation, we produced a range of $Ca^{2+}$ signals in PC dendrites that resembled responses evoked by extrinsic synaptic drive. Low-power light induced relatively small-magnitude $Ca^{2+}$ transients that approximate the $Ca^{2+}$ response elicited by parallel fiber-mediated electrogenic excitation. The more profound depolarization induced by high-power light initiated dendritic action potentials mediated by P/Q-type voltage-gated $Ca^{2+}$ channels, identical to those elicited by climbing fibers[53]. Dendritic spiking elicited by ChR2-mediated depolarization amplified the overall optogenetically induced $Ca^{2+}$ response, greatly increasing its magnitude.

ChR2-evoked $Ca^{2+}$ transients could replace $Ca^{2+}$ signals elicited by either parallel fibers or climbing fibers in inducing plasticity at co-active parallel fiber synapses. While the necessity and sufficiency of PC dendritic $Ca^{2+}$ elevation for inducing LTD are established[37,46,54], our work points to the importance of $Ca^{2+}$-response amplification by dendritic spiking to bias plasticity induction from LTP towards LTD. Hence, our results support the conclusion that the polarity of parallel fiber synaptic plasticity is determined by the magnitude of PC dendritic $Ca^{2+}$ transients with LTD having a higher $Ca^{2+}$ threshold of induction compared to LTP[7]. Blocking $CB_1$ receptors prevented optogenetically induced plasticity, indicating that $Ca^{2+}$-dependent endocannabinoid release from PC dendrites[55] was engaged downstream of optogenetically induced transients and was a requisite component of the instructive signaling pathway in slices. The mechanisms underlying the involvement of endocannabinoid signaling in LTP and LTD at parallel fiber synapses have not been determined[45,46].

**PC dendritic $Ca^{2+}$ signals instruct bidirectional motor learning**. VOR gain-decrease and gain-increase learning resulting from retinal slip errors depend on mechanistically distinct plasticity pathways[56]. Therefore, our central finding that $Ca^{2+}$ signals evoked in PC dendrites are sufficient to instruct bidirectional changes to movement could provide a basis for selecting the appropriate plasticity mechanism to accurately correct erroneous motor behavior across a landscape of different types of mistakes. The gain of the VOR decreased when the vestibular motion was paired with a relatively weak PC optogenetic stimulus. In contrast, the gain of the VOR increased when pairing included a much stronger optogenetic stimulus. Although we were unable to determine whether high-intensity optogenetic PC activation elicited dendritic spiking in vivo to explain the larger evoked $Ca^{2+}$ signals in this stimulus condition, the resulting adapted response resembled that induced by optogenetic activation of climbing fibers, inputs that are known to effectively trigger such electrogenic responses in PCs. Thus, adaptive weakening or strengthening of the VOR is instructed by autonomous PC activity according to a $Ca^{2+}$ threshold rule, similar to plasticity induction at PC synaptic inputs measured in slices. As parallel fiber LTP and LTD are substrates for bidirectional cerebellar motor learning[56,57], our findings support the conclusion that LTP, induced by pairing behavior-evoked parallel fiber activity and small-magnitude PC $Ca^{2+}$ responses, drives an adaptive decrease in VOR gain. In contrast, LTD, induced by pairing behavior-evoked parallel fiber activity and large-magnitude PC $Ca^{2+}$ responses, drives an adaptive increase in VOR gain[56] (Supplementary Fig. 6). Further work will be necessary to determine whether optogenetically induced PC instructive signaling engages the same biological pathways as those engaged during error-driven motor learning. Nonetheless, our results emphasize that $Ca^{2+}$ signaling in PCs is, at least, sufficient for directing bidirectional adaptive changes to motor output.

Interestingly, ChR2-evoked PC $Ca^{2+}$ signals were only effective in instructing changes in VOR gain during particular contexts of vestibular motion. Large-magnitude $Ca^{2+}$ responses induced an increase in VOR gain when they were elicited at the end of ipsiversive, but not contraversive, head turns. This matches the known conditional dependence of climbing-fiber activity that, when optogenetically induced, only elicits VOR gain-increase adaptation when paired with ipsiversive head turns[11,58]. The similar adaptive influence of high-intensity optogenetic PC activation and climbing-fiber stimulation on VOR performance, and the shared conditional effectiveness of their activity at eliciting this learning, further establishes that large-magnitude optogenetically induced dendritic $Ca^{2+}$ transients substitute for instructive signaling mediated by climbing fibers. In contrast, small-magnitude dendritic $Ca^{2+}$ signals induced gain-decrease VOR adaptation when optogenetically elicited at the end of contraversive head turns but not at the end of ipsiversive head turns. Whether small-magnitude $Ca^{2+}$ signals mimic a behavior-mediated response, such as through the activity of parallel fibers or co-activation of climbing fibers and molecular layer interneurons[11,59], is yet to be determined.

The mechanisms underlying the context-dependence of $Ca^{2+}$ signaling for instructing bidirectional motor learning are not clear. However, increasing evidence indicates that plasticity gating, governed by context-dependent neural circuit dynamics, including the engagement of local inhibitory interneuron networks[60,61], may play a widespread role throughout the brain and could therefore contribute to regulating the polarity of plasticity induction and the direction of motor learning in the cerebellum. Certainly, neurons in the vestibulo-cerebellum, including granule cells and molecular layer interneurons, exhibit biased patterns of activity during head turns, with preferences for

either, or both, contraversive or ipsiversive vestibular motion[62–64]. Thus, floccular PCs are likely to integrate different types of extrinsically driven excitatory and inhibitory input, dependent on the context of head turn direction, that could oppose (or favor) the induction of particular forms of plasticity that weaken or strengthen the VOR. Such gating mechanisms may be in place to ensure that only the appropriate form of plasticity is induced to accurately correct the behavior during performance errors that, for the VOR, are referenced to head turn direction.

We purposely timed light pulses at the end of head turns to avoid evoking optogenetic PC $Ca^{2+}$ elevation in combination with behavior-mediated $Ca^{2+}$ responses that are elicited during the peak velocity phase of vestibular motion[11]. Synergist interaction of extrinsic and optogenetically evoked signals could produce behavioral effects distinct from those mediated by either signal source alone. Furthermore, we favored this approach because it allowed optogenetically evoked $Ca^{2+}$ elevation to precisely follow vestibular motion-induced parallel fiber activity, a pairing sequence similar to plasticity experiments performed in slices where evoked $Ca^{2+}$ transients followed parallel fiber tetanus. However, retinal slip reporting VOR performance errors occurs mostly during the peak phase of vestibular motion[65]. Therefore, the timing of our optogenetic stimulus is not reflective of a behaviorally relevant pattern of evoked PC $Ca^{2+}$ activity.

Differences in the timing of optogenetic stimuli during head turns may contribute to apparent disparate findings between our study and a previous report showing gain-increase VOR adaptation, rather than a gain decrease, in response to ChR2-induced PC activation during the peak velocity phase of contraversive vestibular motion[18]. Because this previous study calibrated the intensity of their optogenetic stimulus based on the estimated induced change to PC simple spiking and not the effect on dendritic PC $Ca^{2+}$ signaling, it is difficult to explain the exact mechanisms that contribute to the differences in our results. However, their study used a different Pcp2::Cre driver line[66] in which Cre is expressed in PCs as well as ~20% of molecular layer interneurons[25,67]. Because molecular layer interneuron-mediated inhibition suppresses dendritic spike-evoked $Ca^{2+}$ transients in PCs thus influencing the efficacy and/or polarity of plasticity induction[11], it may be that the coincident optogenetic activation of PCs and molecular layer interneurons alter the direction of VOR adaptation compared to conditions under which PCs are optogenetically activated without the induced activity of MLIs, as in our study, which employed a Pcp2::Cre driver line specific for PCs[25].

**Induction of adaptive behavior involves pathways downstream of PC dendritic $Ca^{2+}$ elevation.** Blocking $CB_1$ receptors prevented both optogenetically induced gain-decrease and gain-increase VOR adaptation. This result matched the endocannabinoid sensitivity of optogenetically induced LTP and LTD induction at parallel fiber synapses in our slice recordings. Thus, convergent plasticity mechanisms involving endocannabinoid-mediated reweighting of parallel fiber synaptic strength likely explains optogenetically induced changes to oculomotor behavior. Although $CB_1$ receptor signaling has been implicated in cerebellar-dependent associative eye-blink conditioning[68,69], a recent investigation using $CB_1$ receptor knockout mice has shown that these receptors are dispensable for learning this task, though the authors could not completely eliminate the possibility of developmental compensation in the absence of their expression[70]. Whether $Ca^{2+}$-dependent retrograde endocannabinoid release from PC dendrites[55,71] is a component of instructive signaling pathways under naturalistic learning conditions, including VOR

adaptation elicited by retinal slip, will thus require future work to untangle. Yet, based on our observation, it is clear that endocannabinoid release from PCs can impart an adaptive influence on motor output, at least when evoked under the artificial condition of optogenetically induced dendritic excitation.

Activating PCs without an accompanying dendritic $Ca^{2+}$ response did not induce a change in VOR gain indicating that optogenetic-evoked simple spiking was not a participant signal contributing to adaptive oculomotor behavior during our training procedure. However, this result does not rule out the possibility that PC simple spiking can instruct aspects of the cerebellar learning process. PC simple spiking modulates during motor performance, encoding information that may be useful for instantiating corrective behavioral modifications during error-driven learning[72,73]. Because PCs alter their spiking dynamics with the acquisition of an adaptive movement, these new patterns could instruct plasticity induction at their postsynaptic target neurons over time[74]. This would allow an initial labile memory of learning, triggered by $Ca^{2+}$-mediated plasticity induction in the cerebellar cortex, to be transferred to a secondary brain site for long-term storage[75]. Such transfer may require hours to days[76], much longer than the time course of our training procedure. Provided our results, future experiments will be required to conclusively show the causal effect of PC simple spiking on imparting motor adaptation separate from the effect of activity perturbations on dendritic $Ca^{2+}$ signaling, which was not parsed in previous studies[17–19].

In summary, our findings establish that PC dendritic $Ca^{2+}$ transients are sufficient, potent triggers of plasticity induction that instruct the acquisition of cerebellar learning. Due to the integrative properties of PC dendrites, extrinsically evoked $Ca^{2+}$ signals vary in magnitude and duration. It is possible that this signal diversity can engage different $Ca^{2+}$-dependent mechanisms that, under naturalistic conditions, contribute to learned changes in movement direction over different time scales[21,77,78]. Such signaling would provide enormous flexibility to the cerebellum in its role in producing appropriate behavioral responses to different adaptive stimuli.

## Methods

**Animals.** All procedures were conducted at the Max Planck Florida Institute for Neuroscience on mature male and female mice (≥8 weeks old for brain slice recording and ≥10 weeks old for behavioral monitoring) using protocols approved by the Institutional Animal Care and Use Committee. Experimental animals were held on a 12-h light/dark cycle in a well-maintained vivarium and had ad libitum access to food and water. We used heterozygous Pcp2::Cre mice (B6.Cg-Tg[Pcp2-Cre]3555jdhu/J]; Jax stock #010536), a driver line with high selectivity for Cre activity in PCs[25,67]. For some experiments, these mice were crossed with the Ai27 reporter line (B6.Cg-Gt[ROSA26]$^{tm27.1-CAG.lsl.ChR2(H134R)-tdTomato}$;/J]; Jax stock #012567), in which loxP-flanked ChR2 was under Cre control resulting in heterozygous offspring that constitutively express the optogenetic actuator across the soma-dendrites of all PCs. In other experiments, Cre-dependent AAV was injected into the cerebellum to transduce PCs with soma-targeted ChR2-$K_v2.1$.

**Brain slice electrophysiology and $Ca^{2+}$ imaging.** To prepare acute brain slices, animals were anesthetized by intraperitoneal injection of ketamine/xylazine (100 and 10 mg/kg, respectively) and then transcardially perfused with cold saline (~4 °C). After their cerebellum was removed by surgical dissection, parasagittal slices (200 μm) were sectioned from the vermis using a vibrating-blade microtome (VT1200S; Leica Biosystems) in an icy solution containing (in mM) 87 NaCl, 25 $NaHO_3$, 2.5 KCl, 1.25 $NaH_2PO_4$, 7 $MgCl_2$, 0.5 $CaCl_2$, 10 glucose, and 75 sucrose. Slices were transferred to an incubation chamber containing bath solution composed of (in mM) 128 NaCl, 26.2 $NaHO_3$, 2.5 KCl, 1 $NaH_2PO_4$, 1.5 $CaCl_2$, 1.5 $MgCl_2$, and 11 glucose. Slices were held in the incubation chamber for 40 min at 34 °C and then at room temperature (23–25 °C) thereafter until use. All solutions were oxygenated with carbogen gas (95% $O_2$, 5% $CO_2$) to equilibrium.

For experiments, cerebellar slices were placed in a submersion chamber under an upright microscope (BX51WI; Olympus) and continuously superfused with warmed bath solution (32–34 °C) containing a $GABA_A$ receptor blocker (100 μM picrotoxin). Where noted, $CB_1$ receptors, voltage-gated $Ca^{2+}$ channels, and/or voltage-gated $Na^+$ channels were blocked by including the following

pharmacological reagents in the bath solution (in µM): 5 AM251, 0.5 ω-agatoxin, 0.5–1 TTX, and 6 mibefradil, respectively (all obtained from Tocris).

PC somata were targeted for whole-cell recording using gradient-contrast microscopy imaging. Recording pipettes (2–6 MΩ) were filled with a filtered intracellular solution containing (in mM): 128 K-gluconate, 2 KCl, 9 HEPES, 4 MgCl$_2$, 4 Na-ATP, 0.5 Na-GTP. In a subset of electrophysiological recordings, PC dendrites were also patched using a fluorescence-guided technique whereby pipettes coated in BSA-conjugated Alexa Fluor-594 (0.02% w/v in 100 µM BSA; ThermoFisher) were guided under continuous two-photon laser-scanning microscopy imaging to the fluorescently labeled neurites of PCs, filled through a somatic patch pipette with volume-indicator dye (40 µM Alexa-594 hydrazide; ThermoFisher). For Ca$^{2+}$ imaging experiments, the green Ca$^{2+}$ indictor dye Fluo-5F was included in the intracellular solution along with the red volume-indicator Alexa-594 hydrazide (200 and 40 µM, respectively; ThermoFisher).

Constant current injection held the resting membrane potential of PCs near −70 mV in current-clamp mode using an ultra-low-noise amplifier (Multiclamp 700B; Molecular Devices). Electrophysiological signals were sampled after low-pass filtering (2–10 kHz) using a digitizer (Digidata 1440 A; Molecular Devices) controlled by acquisition software (pClamp 10; Molecular Devices). Pipette capacitance was neutralized in all recordings and electrode series resistance compensated using the bridge balance circuitry of the amplifier. Two-photon laser-scanning imaging was performed using a custom-built scan head fitted on top of the microscope. A mode-locking Ti:sapphire laser (Chameleon; Coherent) provided excitation light (λ 810 nm); beam steering was accomplished using galvanometer mirrors. Fluorescence emission was chromatically separated using dichroic mirrors and bandpass filters (t560lpxr, and et640/120 or et510/80, respectively; Chroma) and detected using GaAsP photomultiplier modules (Hamamatsu). Calcium imaging data were acquired using the PrairieView software (Bruker).

To cover PC somata and dendritic arbors with light during ex vivo recordings, optogenetic stimuli were delivered by wide-field epi-illumination. This was accomplished by launching light from an unfiltered light-emitting diode (LED) (M470L3; Thorlabs), centered at λ 461 nm (± 20 nm), through the rear epiport of the microscope onto the back of the objective lens. The LED was modulated (<1 kHz) with a current controller (LEDD1B; Thorlabs) triggered using TTL commands produced by the electrophysiology software. For electrical stimuli, brief pulses (100 µs; 1–5 V) were delivered through a constant voltage stimulus isolation unit (DS2A; Digitimer). Climbing fibers were stimulated by placing bipolar glass pipettes placed near the PC soma; parallel fibers were stimulated by placing a separate bipolar glass pipette in the molecular layer near the PC dendritic arbor. The electrical intensity was adjusted such that climbing-fiber stimuli reliably triggered complex spikes in the soma and that parallel fiber stimuli evoked EPSPs 2–4 mV in amplitude. For plasticity experiments, the parallel fiber tetanus consisted of seven closely spaced electrical stimuli (50–100 Hz). This was followed (Δ 25 ms) by either an optogenetic stimulus (one pulse except where noted; 5 ms) or a single climbing-fiber stimulus. The pairing was repeated 30 times over 5 min (0.1 Hz). Test parallel fiber-evoked EPSPs were elicited at 0.05 Hz.

To analyze plasticity experiments, responses were averaged over 30 trials; comparisons were made between a 10 min baseline period immediately prior to the pairing procedure and an epoch 25–35 min after completing pairing. For Ca$^{2+}$ imaging, measurements were made at any location along spiny dendrites. Activity-evoked Ca$^{2+}$ transients were quantified using the ratio of change in the green Ca$^{2+}$ channel divided by the fluorescence signal in the red volume channel (ΔG/R). The optogenetic stimulus artifact was blanked for quantification. To determine the amplitudes of evoked Ca$^{2+}$ responses, we used an exponential fit to the decay of trial-averaged transients. We measured the peak of this fit at the time point immediately after cessation of the optogenetic stimulus or an equivalent time point for climbing-fiber-evoked responses.

**Surgical procedures.** For viral injections, mice were anesthetized by continuous isoflurane gas (1–5%) and held in a stereotactic platform (David Kopf Instruments) by ear bars with thermoregulation provided by a heating plate with biofeedback to maintain body temperature. The anesthetic plane was determined by the absence of toe pinch responses. A subcutaneous injection of lidocaine/bupivacaine was delivered to the scalp. A small incision was opened (<2 mm) allowing for a craniotomy to be cut in the skull (<0.5 µm in diameter). Through this opening, a micropipette containing virus was advanced to the following coordinates (in mm from Bregma): X = ±2.35; Y = −5.65; Z = 3.2–3.4, and α = 10° for targeting the flocculus; X = 0; Y = −6.5; α = 0°; Z = 0.25–1.00 for targeting the vermis. For inferior olive injections, the glass micropipette was instead inserted through the exposed dura mater between the foramen magnum and C1 vertebra at a 62° vertical angle. Bilateral injections were made at 2.6–2.7 mm in depth and 0.24–0.30 mm from the midline. For all injections, undiluted viral solution (titer ≥ 10$^{12}$ vg/mL) was slowly infused into the target site (0.2–0.5 µL per injection). The micropipette was held in place before withdrawal (5–10 min). AAVs included AAV8-EF1α-Flex (*loxP*)rev-ChR2(H134R)-YFP (Addgene #20298), AAV8-EF1α- Flex(*loxP*)rev-ChR2.K$_v$2.1-mRuby, AAV1-CAG-Flex(*loxP*)rev-jRCaMP1a (#AV6587), AAV1-CAG-Flex(*loxP*)rev-ChR2(H134R)-HA-2a-HM4D, and AAV5-CaMKIIa-ChR2 (H134R)-EYFP. Viruses were packaged by the University of Pennsylvania Vector

Core Facility, the University of North Carolina Vector Core Facility, or ViGene Biosciences (Rockville, MD).

For the mice used in behavioral experiments, stainless steel headposts were installed during surgery by removing a section of scalp on the center of the head and attaching it to the exposed skull with dental cement (Metabond; Parkell). Optical fibers (200 µm, NA 0.22 with Ø 1.25 mm ferrules for optogenetics; 400 µm, NA 0.48 with Ø 1.25 mm ferrules for photometry; Thorlabs) were implanted to target the flocculus (coordinates X = ± 3.35 mm; Y = 5.65 mm; α = −14°; Z = 1.9 ± 0.1 mm), secured in place using dental cement. All animals were allowed to recover after surgery under analgesia provided by injection of carprofen and buprenorphine SR-LAB (5 mg/kg and 0.35 mg/kg, respectively). After onset of expression (10–21 days), animals were sacrificed and brains harvested for acute slice preparation or were used for behavioral monitoring. To confirm proper targeting of the flocculus by optical fiber implants in every animal, we measured evoked eye movements to brief optogenetic stimuli during awake quiescence. In rare instances when we could not evoke eye movements, mice were removed from the study, and post hoc inspection of perfused mice revealed improper targeting of ChR2-expressing PCs in the flocculus.

**Behavioral monitoring and training.** Mice were head-restrained using their surgically implanted headposts on a custom-made VOR apparatus. The apparatus included a motorized rotation stage (Zaber Technology) to deliver horizontal vestibular stimuli and a machine-vision camera directed to the left eye so that its position could be tracked in response to passive head turns. Eye movement was determined by pupil position, computed by eye-tracking software (ETL-200; ISCAN). For each animal, we estimated the radius of pupil rotation (Rp) to calculate the angular position of the eye[79]. For this, we measured the pupil position (P) while moving the camera back-and-forth on the rail around the vertical axis of the stage with a known angle (± 10°). The measured P was corrected for motion-induced artifacts by subtracting the corresponding corneal reflection (CR) position coming from an infrared LED fixed to the top of the camera. We used a white-light-emitting LED to modulate pupil diameter by illuminating the eye with various light intensities. Rp was then calculated across pupil diameters according to Rp = Δ/sin (20°). The angular eye position was determined for the corresponding pupil diameter throughout an experiment by the following formula: eye position (Ep) = arcsin [(P$_1$ − CR$_1$)−(P$_2$ − CR$_2$)/Rp]. Prior to experiments, pilocarpine (2% ophthalmic drops; Patterson Veterinary Supply) was briefly applied (<1 min) onto the eye to limit pupil dilatation so that eye movement could be accurately tracked in darkness.

Mice were trained for VOR learning by pairing passive vestibular stimuli with optogenetic activation of PCs (30 min). Sinusoidal stage movements (1 Hz; ±8.5°; 22°/s peak velocity) were used for vestibular stimuli. Light pulses (473 ± 1 nm) from a laser (CNI Optoelectronics Tech; MBL-F-473-200 mW) were used for optogenetic excitation (12 pulses, 5 ms, 50 Hz for PC stimulation; 3 pulses, 20 ms, 8 Hz for climbing-fiber stimulation). Laser light was split into two lines with each line directed through independent acousto-optic modulators (AOMs) (AA Opto-Electronic; MTS110-A3-VIS) allowing for rapid (<1 kHz) modulation of light power of either line. From each AOM, laser light was launched into fiber ports (PAF-X-11-A; Thorlabs); patch cables then delivered light (0.5–20 mW out of patch cable) to the optical fiber implants targeting the flocculi of experimental animals. Light pulses were triggered by stage position such that optogenetic activity was induced relative to the head.

Immediately prior to training, a test measurement was obtained to gauge baseline VOR performance. After training, the VOR was re-tested. VOR gain changes were computed as the percentage difference in performance after training relative to the baseline measurement before training (ΔVOR). This normalization procedure facilitated comparison across sessions and between mice. In addition to sessions employing optogenetic stimuli, measurement of VOR gain was also made in separate sessions that included training with the vestibular stimulus alone. This control condition fully accounted for darkness-induced habituation. For each mouse, the order of training sessions was randomized with at least two days between tests for all conditions. Baseline VOR performance across sessions was not significantly different (ANOVA with Bonferroni post hoc tests; P > 0.05).

For photometry, light centered on λ 565 nm (~10–40 µW) from a fiber-coupled LED (M565F3; Thorlabs) was launched into a fiber-compatible dichroic mirror mount (Fluorescence Mini-cube; Doric Lenses) that combined blue laser light for optogenetic excitation. A patch cable from the fiber port delivered excitation light to implanted optical fibers. Current controllers (LEDD1B; Thorlabs) triggered by TTLs were used to toggle the LED. Emitted fluorescence was collected through the same implanted optical fiber, passed back through the dichroic mirror mount, and detected using a femtowatt, visible wavelength photoreceiver (Model 2151; Newport) at a high sampling rate (2 kHz). Data points were averaged, producing an effective rate of activity measurements of 25 Hz.

**Histology and fluorescence microscopy.** For post hoc examination of ChR2 expression and optical fiber-placement confirmation, mice were deeply anesthetized by intraperitoneal injection of ketamine/xylazine (100 and 10 mg/kg, respectively) and then transcardially perfused with cold tris-buffered saline (TBS) followed by 4% paraformaldehyde (PFA) in TBS. After overnight post-fixation in PFA, the cerebellum was removed by dissection and cut into thin sections

(60–80 μm) that were mounted onto glass slides. To immunohistochemically label PCs, cerebellar slices were incubated for one hour at room temperature in blocking solution (10% normal goat serum and 0.2% Titron X-100 in TBS) and then overnight in primary antibodies (anti-calbindin #CB38a, Swant; 1:1000 in 5% normal goat serum 0.1% Triton X-100 in 1× TBS) at 4 °C. After washing in phosphate-buffered saline, slices were incubated for 1 h at room temperature in secondary antibodies, either Alexa Fluor-633 goat anti-rabbit (1:1000 #A-21070, Thermofisher) or Alexa Fluor-488 goat anti-rabbit (1:1000, #A-27034, Thermofisher), and then mounted on slides after repeat washes in TBS.

**Data analysis, statistics, and reproducibility**. Matlab (Mathworks), Axograph X (Axograph), and ImageJ (NIH) were used for data analysis. The Matlab code is available from the corresponding author upon request. Graphpad Prism (Graphpad Software) was used for statistical analysis with values in the text and figures reported as mean ± SEM; a summary table includes a description of these tests with replicate values (Supplementary Table 1). In short, differences between groups of data were deemed significant with $\alpha$ values of $P < 0.05$. All tests were two-sided. Paired or unpaired $t$ tests were used to compare two groups depending on whether comparing matched or unmatched values. For plasticity experiments, because normal distribution cannot be assumed, nonparametric Wilcoxon tests were used to compare matched values before and after plasticity induction. When comparing more than two conditions or factors, group data were compared with one or two-way ANOVA and significance between groups was determined with post hoc tests. Tukey's post hoc test was used for behavioral experiments to compare optogenetically stimulated groups with the vestibular stimulation-only control group. Dunnett's post hoc comparison test was used to compare each data set with every other condition. Sidak's post hoc test was used to compare pairs of means in two different conditions (AM251 behavioral experiment).

The expression profile of fluorescent proteins in cerebellar cells shown in representative images in Fig. 6d and Supplementary Figs. 2b and 5a was observed in at least three animals, for Figs. 1a and 5a, it was observed in more than ten animals.

**Reporting summary**. Further information on research design is available in the Nature Research Reporting Summary linked to this article.

## Data availability
The data that support this study's findings are available from the corresponding author upon request. Source data are provided as a Source Data file.

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

## Acknowledgements

This work was supported by the National Institutes of Health Grants NS105958 and NS112289 (J.M.C.), the Max Planck Society (Max Planck Gesellschaft), and the Max Planck Florida Institute for Neuroscience. We thank all members of the Christie Lab for their helpful discussions and comments during the completion of this project. We also thank the GENIE project at the Janelia Research Campus for making their Ca²⁺ sensors available for public use.

## Author contributions

Conceptualization: A.B. and J.M.C.; methodology: A.B. and J.M.C.; reagents: C.A.B. and M.M.B.; investigation: A.B. and M.J.R.; writing and editing, A.B. and J.M.C.; funding acquisition: J.M.C.; supervision: J.M.C.

## Funding

## Competing interests

The authors declare no competing interests.
