## [Peer Review File · Nature Communications]

Reviewer #1 (Remarks to the Author):

This is an interesting and comprehensive paper that has addressed the question of whether the subcellular location of a calcium signal in Purkinje cells – effectively, a global dendritic calcium signal that is analogous to the climbing fiber signal – is sufficient to drive plasticity and behavior. The authors do this by utilizing cell-specific expression of a ChR2 that is particularly highly expressed in dendrites, and applying either a low light or high light to elicit sub- or suprathreshold signals for dendritic calcium spikes. The authors then show that suprathreshold optogenetic activation can be used to produce PF-LTD and subthreshold activation can be used to produce PF-LTP. They then show that similar protocols in vivo can produce changes in VOR adaptation. Finally, they show that these VOR adaptations (and LTP and LTD) require endocannabinoid signaling, and that optogenetic activation without dendritic calcium influx is insufficient to lead to changes in VOR.

This paper adds something interesting to the cerebellar field by providing clear evidence of the role of dendritic calcium spikes in LTD but not LTP and is commendable for its effort to translate this to the intact behaving animal. Technically, this paper is a tour de force, utilizing multiple techniques including pushing the envelope of what is possible utilizing optogenetics, as well as dendritic recording, plasticity experiment, pharmacology, and mouse behavior. Thus, this paper will be of interest to many in the community, and will influence thinking in the field.

Major Comments

1. Why do adaptations occurs only for contralateral stimulation for low intensity and ipsilateral for high intensity? This finding is puzzling, and makes the paper hard to integrate. While I do not think that further experiments are warranted, I do think that more substantive discussion of this is at present missing.
2. Similarly, contextualizing the findings of this study with others in the field (e.g. Nguyen-Vu et al., Boyden et al.) is not fully convincing, and should be addressed more directly. You say you have expression in 20% of interneurons, but that is not seen in your example images. Can you show images that highlight this?

Minor Comments

1. Dashed lines showing pre-training VOR are so faint that they are almost invisible. Similarly, the light blue text for “low intensity” light stimulation is too faint and difficult to read in Fig. 6, and Supplementary Fig. 3
2. The authors claim that ChR2 is well-expressed throughout the cell, but the staining in the soma appears extremely low, and thus this statement should be updated to reflect that.
3. Fig. 2. The color used for both Control and Climbing fiber stimulation is the same in panels b and c, which is confusing.
4. Supplementary Fig. 3: The axis appear flipped, since Ipsi stimulation appears to cause positive displacement (panel a) yet is negative in panel c, and vice vesa for Contra stim.
5. Page 6 typo: “with the different polarities of alteration have varying Ca²⁺ thresholds of induction” should be “having”
6. Figure 2 legend: “Test EPSP amplitude was no affected” should be “not”.

Reviewer #2 (Remarks to the Author):

In this manuscript, Bonnan et al. use optogenetic activation of Purkinje cells to investigate the strength and subcellular localization of calcium-currents necessary for opposite-polarity plasticity events, LTP and LTD. They show that the subthreshold stimulation induces low calcium currents, without dendritic spiking, that induce LTP when paired with PF stimulation. In contrast, suprathreshold activation causes high calcium currents, with dendritic spiking, and LTD when paired with PF stimulation. Next, they show that they can induce VOR motor learning using their stimulation paradigms in vivo, with low power contraversive stimulation causing increased gain and high power ipsiversive stimulation causing decreased gain. Finally, they show that this gain change is dependent on endocannabinoid signaling (that the authors presume is activated through retrograde signaling in the dendrites) and on dendritic activation. Altogether the authors present an impressive amount of work, the figures are clear, and the text is generally well written. However, despite its many merits, there are some overarching issues that are confusing and require attention from the authors.

General comments:

- 1) As one reads the paper, it quickly becomes unclear what specific question the authors are trying to address from this work and how the results precisely expand upon or connect existing theories in the literature. Please see below for additional comments, but I recommend focussing the argument.
- 2) Related to above, the title is not particularly informative. Especially since none of the in vivo experiments show a graded response, thus the authors really do not "deconstruct" any "motor learning". Maybe they do "deconstruct" ex vivo opposite-polarity plasticity that was largely known, which is an argument I am basing on from what this manuscript has cited in the text. I would strongly suggest revising the title to be more descriptive, and also a better argument/question needs to be built out in the introduction, especially since the data in the Results section have been written and described for a specific audience, namely the experts in that field.
- 3) In some cases the interpretations are not clear. For instance, in Figures 3e and h, the black lines are the VOR gain. Based on these panels, the authors claim a difference in "low intensity" and "high intensity". However, unless I am missing it, the authors never do direct statistical comparisons between the two experimental paradigms. There is something confusing about how the data are presented.
- 4) From a very broad perspective, it is not clear to me how physiologically relevant the optogenetic activation actually is, specifically as it relates to the goal of this study. That is, what is the evidence to show that the calcium transients that are observed upon optogenetic activation actually induce the same biological pathways as more natural behaviorally related Purkinje cell responses in vivo? In this regard, a deeper discussion about the methodology is warranted.

Specific comments:

- 1) The authors postulate that dendritic firing events occur during LTD, but not LTP. They base this conclusion on their finding that low intensity stimulation does not cause dendritic firing in ex vivo preparations and that low intensity stimulation induces LTP. This is an interesting hypothesis. Yet, the in vivo experiments presented in this paper may not entirely support this hypothesis. First, in figure 4, the authors show that LTP is dependent on endocannabinoid signaling, which they postulate occurs through dendritic calcium activity. Second, in Figure 6 they show that activation of the Purkinje cell body (but not dendrites) is insufficient to increase VOR gain. While all the ex vivo data are supportive of the notion that relatively lower calcium currents and PC activation results in LTP, whereas stronger calcium currents result in LTD, the authors do not provide sufficient information to support their hypothesis that dendritic spiking explains the difference in synaptic

plasticity. Furthermore, in Figures 4-6, the authors do not find any differences in mechanistic dependence on the specific manipulations between low- and high- intensity stimulation paradigms, which leaves the question whether the in vivo activation paradigms are indeed different from each other. Apologies if I have missed something here, but the authors present a great many pieces of data and the links between them a sometimes challenging to appreciate.

2) One interesting result is the directional dependence of PC excitation during gain changes, (ipsiversive motion paired with high-intensity and contraversive motion paired with low intensity stimulation). The authors suggest that these results are due to contextual difference. However, when comparing Figure 3e with 3h, it seems that the most apparent difference is in the “Vestib.”/black lines. In the low intensity paradigm, the Vestib. Group has a VOR gain of ~25%, whereas in the high intensity paradigm, the Vestib. Group has a VOR gain of ~40%. The Ipsi./blue group has a similar VOR gain in both stimulation paradigms (~25%) and so does the Contra./red group (~40%). The significant results could therefore be driven by the difference in intrinsic gain in the unstimulated trials rather than a mechanistic difference between low- and high- stimulation paradigms. Furthermore, the low- and high- stimulation paradigms are never directly compared to each other (if they would be, the “Vestib” trials would likely be the only significant difference). In other words, what is driving the different between experimental conditions?

3) Similarly, when comparing 4e to 4f, the Vestib/black results change in opposite direction of the measured effects of contra/ipsi stimulation (this figure also lacks the direct comparison between ipsiversive and contraversive stimulation in either high or low intensity stimulation groups). What is driving the large variability in “Vestib.” Gain changes between animals in the different experimental setups (compare “Vestib.” VOR gain between figures 3, 4, 6, and supplemental figure 4)? And does ipsiversive stimulation always result in relatively lower gain changes than contraversive stimulation when doing within animal comparisons?

4) VOR changes also occur in the unstimulated “Vestib.” Group, suggesting that some naturally occurring events still drive plasticity in this group. How does this plasticity interfere with the plasticity induced by optogenetic stimulation? This question is specifically relevant in the context of the difference in results obtained this study and the cited literature (DOI: 10.1038/nn.3576).

5) For supplemental figure 4 and figure 4, why were the stimulation experiments only performed in one direction?

6) Figure 4: how does AM251 effect “natural” learning events through climbing fiber stimulation? This control is essential to show that the effect of endocannabinoid signaling is a natural occurring event and not only present during the optogenetic induced plasticity.

7) Figure 5b: the order of the example traces (left) is different from the order of the bars (right). For clarity and simplification, it would be beneficial to be consistent with the order.

8) Figure 5f: Please clarify what the control experiment is. And, can climbing fiber stimulation in mice injected with these viral constructs still induce LTD?

9) Figure 6d should be discussed before Figure 6f as it is presented as such in the figure (or the order of the text should be reversed).

- 10) For the ex vivo recordings, what does the n-number represent? Number of mice? Cells? If cells, from how many mice did these cells come?
- 11) Methods: Can the authors please clarify when they used the different post-hoc tests for the ANOVA?
- 12) The authors are presenting a lot of experimental manipulations and results. It would be beneficial for readers to provide a summary table with all the results that allow for direct comparison between the different conditions. Alternatively, a summary figure with the proposed mechanism would be beneficial to readers that are less familiar with this field.
- 13) Number of mice used per experiment/ information about experimental replication is lacking throughout the paper. Additionally, in the cases when "n" is present, it is not clearly defined.
- 14) The leaky expression of Pcp2 in the molecular layer interneurons is worrying. Please confirm that there was no ChR2 expression in climbing fibers, mossy fibers, or parallel fibers through the relevant regions of the cerebellum. Higher power images should be shown.
- 15) Successful targeting of optical fibers must be demonstrated.
- 16) Please clarify where in the cerebellum the slice recordings and Ca²⁺ imaging of Purkinje cells were performed and justify why this location was chosen.
- 17) Please provide further discussion of what may be causing the increased Ca²⁺ when a greater light power is used. Do you believe this is activation of greater number of ChR2 channels or some other mechanism?
- 18) I would recommend providing the mean and SEM for the statement: "the change in synaptic efficacy obtained by conjunctive pairing with climbing fibers was greater than that obtained by optogenetically induced PC activation (p = 0.04, t-test)."
- 19) The colors of the dotted lines in panels 3d and 3g are too faint to be appreciated. Please saturate the colors further.
- 20) Please state the statistical values of panels 4e and 4f in the main body of the text.
- 21) The authors mention on page 11 adding more optogenetic stimulation pulses in order to "increase the overall level of optogenetically induced excitation." Please provide further detail in the text and figures to how many different levels of optogenetic stimulation were tried, the nature of the stimulus pattern, and the results of each attempt. How did the number and timing of spikes elicited in the soma-targeted stimulations compare to the whole-cell expression of ChR2?
- 22) The statement: "Over the time course of training, optogenetically induced simple spiking in PCs was insufficient to elicit behavioral change," is a bit vague and confusing. Is this meant to only reference the soma-targeted ChR2 or was there generally no obvious change in behavior?

23) The explanation on page 15 of the different findings of this work and that of Nguyen-Vu, et al. needs expansion. How does this manuscript's finding of a lack of effect after ipsiversive training in the low-intensity condition compare to the author's suggestion that Nguyen-Vu et al. initiated LTP? What of the differences between this manuscript's and Nguyen-Vu et al.'s climbing fiber stimulation experiments?

24) Page 23: I would recommend reporting drug dosages in mg/kg.

25) Is a period missing at the end of the sentence in figure legend for panel 6d or is there information that is cut off?

Reviewer #3 (Remarks to the Author):

The manuscript by Bonnan et al is a tour de force examination of the Purkinje cell dendritic calcium signal amplitudes that drive bidirectional parallel fiber plasticity in the cerebellar cortex. The authors take advantage of optogenetics to drive changes in the dendritic calcium in Purkinje cells, and “calibrate” the stimulations to generate calcium signals which span from sub to suprathreshold in comparison to activation of climbing fibers. The authors show, quite convincingly, that the calcium rise in the dendrites of Purkinje cells can bidirectionally produce synaptic plasticity. Then they demonstrate that by titrating the dendritic calcium changes they can produce VOR gain changes. These changes are not brought about by a change in simple spike firing rate, but require changes in dendritic calcium because when ChRd expression was restricted to the soma of Purkinje cells it failed to induce plasticity. Lastly, and for me perhaps a bit of distraction, the authors show that plasticity in vivo is dependent on endocannabinoid signaling.

Overall, I think this is an outstanding paper. The manuscript represents a rigorous set of experiments that have been expertly conducted, and clarifies an important question in the field, and in my opinion rights many published wrongs. I frankly struggled to identify issues that I had concerns about. There are two exceptions to this. Megan Carey’s lab has presented a body of work which suggests that the CB1 receptors are not, as suggested before, required for learning in vivo if the KO mice are made to walk at the same pace as the wild type mice. That story is at odds with the results reported here. The second minor concern is that the calcium signals seems to have gotten bigger when voltage gated Na channels were blocked with TTX (supl figure 1), and I cannot understand what the cause of this increase could be.

Overall, this paper would be an invaluable contribution to the field, and I congratulate the authors for a job well done.

Kamran

Below are our responses to the reviewers' concerns. We indicate in bold our changes to the manuscript (page number corresponds to the version of the revised manuscript with tracked changes).

Reviewer #1:

“This is an interesting and comprehensive paper that has addressed the question of whether the subcellular location of a calcium signal in Purkinje cells – effectively, a global dendritic calcium signal that is analogous to the climbing fiber signal – is sufficient to drive plasticity and behavior. The authors do this by utilizing cell-specific expression of a ChR2 that is particularly highly expressed in dendrites, and applying either a low light or high light to elicit sub- or suprathreshold signals for dendritic calcium spikes. The authors then show that suprathreshold optogenetic activation can be used to produce PF-LTD and subthreshold activation can be used to produce PF-LTP. They then show that similar protocols in vivo can produce changes in VOR adaptation. Finally, they show that these VOR adaptations (and LTP and LTD) require endocannabinoid signaling, and that optogenetic activation without dendritic calcium influx is insufficient to lead to changes in VOR.

This paper adds something interesting to the cerebellar field by providing clear evidence of the role of dendritic calcium spikes in LTD but not LTP and is commendable for its effort to translate this to the intact behaving animal. Technically, this paper is a tour de force, utilizing multiple techniques including pushing the envelope of what is possible utilizing optogenetics, as well as dendritic recording, plasticity experiment, pharmacology, and mouse behavior. Thus, this paper will be of interest to many in the community, and will influence thinking in the field.”

We thank the reviewer for finding our work interesting and concluding that it will have a significant impact on the field. We also appreciate the recognition that our use of cutting-edge technology enables us to link cellular-level processing to behavior. We believe there is a dearth of knowledge addressing this important problem and this missing information precludes a greater understanding of the neural circuit processes in the cerebellum that underlie its function in motor learning and memory.

“Why do adaptations occur only for contralateral stimulation for low intensity and ipsilateral for high intensity? This finding is puzzling, and makes the paper hard to integrate. While I do not think that further experiments are warranted, I do think that more substantive discussion of this is at present missing.”

In short, we do not have a mechanistic explanation as to why different intensities of PC activation instruct disparate adaptive effects on VOR performance. One possibility is that the behavior drives different patterns of extrinsic activity onto PCs (i.e., the integrated activity of different populations of parallel fibers, molecular layer interneurons [MLIs], and other PCs), tuned to the direction of head motion. These different patterns of activity could then differentially alter (or “gate”) the susceptibility to different types of plasticity induction in a context-dependent manner. Inhibition from interneurons is a common gating mechanism throughout neural circuits in the brain. We have previously shown that inhibition from MLIs in the cerebellum can gate plasticity induction (Rowan et al. 2018) so perhaps this cell type plays an outsized role in regulating plasticity induction during behavior. To gain insight into this possibility, it would require extensive knowledge of how the ensemble of neurons in the flocculus (e.g., MLIs) activate across VOR contexts and ascertaining whether the behavior drives distinct types of Ca²⁺ signals in PCs during these contexts and the resulting effect of MLI activity on the evoked responses. Currently there is the near absence of such measurements from mice. **To address this concern, we have added to the discussion in an effort to assist readers in integrating our findings (pp. 16 and 17).**

“Similarly, contextualizing the findings of this study with others in the field (e.g. Nguyen-Vu et al., Boyden et al.) is not fully convincing, and should be addressed more directly. You say you have expression in 20% of interneurons, but that is not seen in your example images. Can you show images that highlight this?”

To be clear, we did not use a mouse line with off-target Cre expression in interneurons. On the contrary, we chose a line with Cre expression only in PCs (the *Pcp2::Cre^{idhu}* line). The reason for the reviewer's confusion was a poor choice of wording on our part. Specifically, our statement: "...this study used a *Pcp2::Cre* driver line..." (p. 15) was in reference to Nguyen-Vu et al. 2013, whom we cited in the previous sentence. In their study, Nguyen-Vu et al. used *Pcp2::Cre^{mpin}* mice, a driver line with Cre expression in MLIs (see Witter et al., 2016). Because we have previously shown that optogenetic-induced MLI activity influences VOR learning (Rowan et al. 2018), we believe that off-target optogenetic stimulation of ChR2-expressing MLIs in *Pcp2::Cre^{mpin}* mice is problematical and could lead to confounding results. As presented in our response to Reviewer 2, we conclusively show that differences between our study and Nguyen-Vu et al. are, in fact, attributable to the mouse strain used. **To address the reviewer's concern, we edited our wording (p. 16). As mentioned above, we also have tried to contextualize our findings with previous studies to make these comparisons more convincing (pp. 3, and 16-18).**

"Dashed lines showing pre-training VOR are so faint that they are almost invisible. Similarly, the light blue text for "low intensity" light stimulation is too faint and difficult to read in Fig. 6, and Supplementary Fig. 3"

We edited the lines in these figures (Fig. 6 and Supplementary Fig. 3) to make them more visible.

"The authors claim that ChR2 is well-expressed throughout the cell, but the staining in the soma appears extremely low, and thus this statement should be updated to reflect that.

We changed the text to address this comment (p. 5).

"Fig. 2. The color used for both Control and Climbing fiber stimulation is the same in panels b and c, which is confusing."

We edited the figure (Fig. 2) to address this comment.

"Supplementary Fig. 3: The axis appear flipped, since Ipsi stimulation appears to cause positive displacement (panel a) yet is negative in panel c, and vice versa for Contra stim.

The reviewer is correct. We had inadvertently flipped the graph's axis. We apologize for this error. The updated figure (Supplementary Fig. 3) is now correct.

"Page 6 typo: "with the different polarities of alteration have varying Ca²⁺ thresholds of induction" should be "having" and "Figure 2 legend: "Test EPSP amplitude was no affected" should be "not".

We corrected these typos (pp. 6 and 35).

Reviewer #2:

"In this manuscript, Bonnan et al. use optogenetic activation of Purkinje cells to investigate the strength and subcellular localization of calcium-currents necessary for opposite-polarity plasticity events, LTP and LTD. They show that the subthreshold stimulation induces low calcium currents, without dendritic spiking, that induce LTP when paired with PF stimulation. In contrast, suprathreshold activation causes high calcium currents, with dendritic spiking, and LTD when paired with PF stimulation. Next, they show that they can induce VOR motor learning using their stimulation paradigms in vivo, with low power contraversive stimulation causing increased gain and high power ipsiversive stimulation causing decreased gain. Finally, they show that this gain change is dependent on endocannabinoid signaling (that the authors presume is activated through retrograde signaling in the dendrites) and on dendritic activation. Altogether the authors present an impressive amount of work, the figures are clear, and the text is generally well

written. However, despite its many merits, there are some overarching issues that are confusing and require attention from the authors.”

“As one reads the paper, it quickly becomes unclear what specific question the authors are trying to address from this work and how the results precisely expand upon or connect existing theories in the literature. Please see below for additional comments, but I recommend focussing the argument.”

“Related to above, the title is not particularly informative. Especially since none of the *in vivo* experiments show a graded response, thus the authors really do not “deconstruct” any “motor learning”. Maybe they do “deconstruct” *ex vivo* opposite-polarity plasticity that was largely known, which is an argument I am basing on from what this manuscript has cited in the text. I would strongly suggest revising the title to be more descriptive, and also a better argument/question needs to be built out in the introduction, especially since the data in the Results section have been written and described for a specific audience, namely the experts in that field.”

We agree that our title lacked specificity. **Therefore, we have retitled the manuscript to make it more informative.** We disagree that our work is somehow unfocused. Specifically, we are addressing an important question in the cerebellar field: what are the signals available to PCs that allow them to encode a diverse range of adaptive behavioral responses. Our results using optogenetics to drive cell-autonomous PC activation sheds light on this question and provides insight into the role of dendritic Ca^{2+} signaling in imparting bi-directional adaptive control over movement strength. Other recent investigations, published in high-profile journals, have employed PC-autonomous optogenetic stimulation to address this question as well. However, our results raise serious concerns regarding how well controlled these previous investigations were conducted and thus their overall conclusions. That said, to address the reviewer’s concern, **we have added text to the manuscript to connect our results with prior studies as well as existing theories about cerebellar learning (pp. 3 and 16-18).**

“In some cases the interpretations are not clear. For instance, in Figures 3e and h, the black lines are the VOR gain. Based on these panels, the authors claim a difference in “low intensity” and “high intensity”. However, unless I am missing it, the authors never do direct statistical comparisons between the two experimental paradigms. There is something confusing about how the data are presented.”

We performed the requested comparison and found a significant difference between the two experimental conditions (p. 9). We present this data in new Supplementary Fig. 4. For this, we measured the ΔVOR at time-matched points in the same mice for test training sessions incorporating pairing of PC optogenetic activation with vestibular stimulation as well as in control sessions consisting of only the vestibular stimulus. Then we subtracted the darkness-induced change determined in the control session from the change measured in the test sessions. This normalization procedure, used in other studies (Nguyen-Vu et al., 2013; Kimpo et al., 2014; Rowan et al., 2018), accounts for between-animal variability stemming from differences in their intrinsic baseline VOR set point as well as their individual sensitivities for darkness-induced habituation. This was necessary for this requested comparison because we used partially overlapping sets of mice when testing the adapting effect of low- and high-intensity PC activation on VOR gain. Without this normalization procedure, these sources of variability could obscure learning-related differences between groups of mice. In this respect, the subtracted difference between conditions is more relevant than the absolute values for this comparison.

“One interesting result is the directional dependence of PC excitation during gain changes, (*ipsiversive* motion paired with high-intensity and *contraversive* motion paired with low intensity stimulation). The authors suggest that these results are due to contextual difference. However, when comparing Figure 3e with 3h, it seems that the most apparent difference is in the “Vestib.”/black lines. In the low intensity paradigm, the Vestib. Group has a VOR gain of ~25%, whereas in the high intensity paradigm, the Vestib. Group has a VOR gain of ~40%. The Ipsi./blue group has a similar VOR gain in both stimulation paradigms

(~25%) and so does the Contra./red group (~40%). The significant results could therefore be driven by the difference in intrinsic gain in the unstimulated trials rather than a mechanistic difference between low- and high- stimulation paradigms. Furthermore, the low- and high- stimulation paradigms are never directly compared to each other (if they would be, the “Vestib” trials would likely be the only significant difference). In other words, what is driving the different between experimental conditions?

As mentioned in our response above, for these experiments, each animal was tested in a control training session as well as in training sessions using at least one level of optogenetic-induced PC activity (i.e., pairing vestibular motion with either low-intensity or high-intensity stimulation). To determine the adaptive effect of optogenetic PC activation on VOR performance, we compared the baseline-subtracted change in VOR gain (Δ VOR; this normalizes for between-animal variability in the intrinsic set point of their VOR) in the test condition relative to that observed for the same mouse in the control training session. This previously established approach accounts for darkness-induced habituation that varies between animals. Using this approach, we could exclude the possibility that intrinsic gain differences and/or sensitivity to darkness-induced habituation between animals underlies differences resulting from the optogenetic pairing procedure. As mentioned above, we also performed the requested comparison between the low- and high-intensity conditions.

“Similarly, when comparing 4e to 4f, the Vestib/black results change in opposite direction of the measured effects of contra/ipsi stimulation (this figure also lacks the direct comparison between ipsiversive and contraversive stimulation in either high or low intensity stimulation groups). What is driving the large variability in “Vestib.” Gain changes between animals in the different experimental setups (compare “Vestib.” VOR gain between figures 3, 4, 6, and supplemental figure 4)? And does ipsiversive stimulation always result in relatively lower gain changes than contraversive stimulation when doing within animal comparisons?”

As mentioned in our responses above, the intrinsic set point of the VOR varies between animals. This contributes to inter-group variability in the control conditions because we used partially overlapping sets of mice when testing for the adapting effect of the low- and high-intensity optogenetic stimulus conditions.

“VOR changes also occur in the unstimulated “Vestib.” Group, suggesting that some naturally occurring events still drive plasticity in this group. How does this plasticity interfere with the plasticity induced by optogenetic stimulation? This question is specifically relevant in the context of the difference in results obtained this study and the cited literature.”

The decrease in VOR gain in the unstimulated control group results from darkness-induced habituation (Stahl, 2004; Nguyen-Vu et al., 2013; Kimpo et al., 2014; Voges et al., 2017; Rowan et al., 2018). Habituation is short-lived phenomenon that rapidly reverses when the animal returns to a lighted environment. It is unclear where habituation originates and may result from neural activity outside of the cerebellum. There is no evidence that it interferes with plasticity driven by the optogenetic stimulus. We use a standard approach to reveal learning apart from habituation. As mentioned in our responses above, this includes the use of a control training session without optogenetic pairing as reference to compare against the changes induced by the pairing procedure in test training sessions.

“From a very broad perspective, it is not clear to me how physiologically relevant the optogenetic activation actually is, specifically as it relates to the goal of this study. That is, what is the evidence to show that the calcium transients that are observed upon optogenetic activation actually induce the same biological pathways as more natural behaviorally related Purkinje cell responses in vivo? In this regard, a deeper discussion about the methodology is warranted.”

Our investigation tested the sufficiency of autonomous PC activation to induce different types of adaptive behavior. We were excited to find that we could induce adaptive responses that were convergent with error-

driven VOR learning. Whether VOR gain weakened or strengthened depended on the intensity of PC activation suggesting that opposite-direction forms of learning were engaged in a mechanistically distinct manner. As the reviewer points out, it is unclear whether optogenetic-induced PC activation actually engages the same biological pathways as those involved in encoding error-driven motor learning. This is difficult to test directly because the key biological pathways that underlie error-driven learning in the cerebellum have not been determined (there are numerous conflicting reports). In this regard, further investigation will be required to determine exactly how PCs in the murine flocculus are activated when the VOR is performed in different contexts (i.e., with and without different directions of retinal slip), including behavior-evoked Ca^{2+} signals in their dendrites, and how this activity engages distinct biochemical pathways to induce opposite forms of plasticity that encode a range of learning outcomes in response to different contexts of mistakes. **To address the reviewer's concern we have edited the manuscript to point out that there are important caveats in interpreting the relevance of optogenetic-induced adaptation in comparison to learning driven by motor errors (pp. 16-18).**

“The authors postulate that dendritic firing events occur during LTD, but not LTP. They base this conclusion on their finding that low intensity stimulation does not cause dendritic firing in ex vivo preparations and that low intensity stimulation induces LTP. This is an interesting hypothesis. Yet, the in vivo experiments presented in this paper may not entirely support this hypothesis. First, in figure 4, the authors show that LTP is dependent on endocannabinoid signaling, which they postulate occurs through dendritic calcium activity. Second, in Figure 6 they show that activation of the Purkinje cell body (but not dendrites) is insufficient to increase VOR gain. While all the ex vivo data are supportive of the notion that relatively lower calcium currents and PC activation results in LTP, whereas stronger calcium currents result in LTD, the authors do not provide sufficient information to support their hypothesis that dendritic spiking explains the difference in synaptic plasticity. Furthermore, in Figures 4-6, the authors do not find any differences in mechanistic dependence on the specific manipulations between low- and high- intensity stimulation paradigms, which leaves the question whether the in vivo activation paradigms are indeed different from each other. Apologies if I have missed something here, but the authors present a great many pieces of data and the links between them a sometimes challenging to appreciate.”

The reviewer is correct; we did not record subcellular Ca^{2+} -spike activity in PC dendrites in response to optogenetic stimulation during behavior. This is obviously a very challenging experiment, though one we would like to attempt in the future. Importantly, using *in vivo* photometry measurements, we determined that the level of optogenetic-induced PC Ca^{2+} activity increases with the intensity of the optogenetic stimulus. Previous work established that the sign of parallel-fiber-to-PC plasticity is determined in a Ca^{2+} dependent manner with LTD having a greater induction threshold (Coemans et al., 2004). Therefore, it stands to reason that high-intensity optogenetic stimuli, which drives the largest level of Ca^{2+} activity and induces VOR gain-increase adaptation, is sufficient to induce LTD as we observe in our *ex vivo* slice experiments. Notably, these plasticity results match prior reports showing that both LTP and LTD are sensitive to endocannabinoid receptor blockade (Safo et al., 2005; Wang et al. 2014). It is unclear how signaling through this common pathway leads to disparate plasticity types at parallel fiber synapses. **To address the reviewer's concern, we have edited the manuscript specifying the limits of our experimental approach to measure PC Ca^{2+} activity *in vivo* and have tempered our conclusions accordingly (p. 15).**

“Figure 4: how does AM251 effect “natural” learning events through climbing fiber stimulation? This control is essential to show that the effect of endocannabinoid signaling is a natural occurring event and not only present during the optogenetic induced plasticity.”

We attempted to address this question by injecting the CB1 receptor antagonist AM251 into wildtype mice and measuring their ability to adapt their VOR in response to opposite-direction visual-vestibular mismatch training. However, we found that injections of DMSO alone (the solvent used for AM251 injection solutions) inhibited gain-increase learning (see the figure on the right). This confounding result prevents any meaningful interpretation of the effect of AM251 on error-driven VOR learning. Importantly, DMSO does not impair VOR adaptation induced by optogenetic PC activation (Fig. 4e.4f) indicating that the obfuscating effect of DMSO on error-driven VOR learning occurs upstream of PC instructive signaling. Notably, Megan Carey's group just reported a similar obfuscating effect of DMSO on eye blink conditioning, preventing them from using pharmacology to directly test the role of CB1 receptors in this cerebellar-dependent learning task (Albergaria et al., 2020). Although the Carey group went on to use CB1 receptor KO mice to show that CB1 receptors are dispensable for the acquisition of this learned behavior, they could not completely eliminate the possibility of developmental compensation in the absence of CB1 receptor expression (hence, their attempt to use pharmacology to address this concern). In light of these observations, it is clear that a dedicated investigation will be required to fully test the role of CB1 receptors in cerebellar learning and thus is beyond the scope of our current study. **In response to the reviewer's concern, we have cited Albergaria et al., 2020 and clarified that there are important caveats to consider when interpreting our results, specifically regarding of the role of CB1 receptor signaling outside of learning induced by optogenetic PC activation (p. 18).**

“The leaky expression of Pcp2 in the molecular layer interneurons is worrying. Please confirm that there was no ChR2 expression in climbing fibers, mossy fibers, or parallel fibers through the relevant regions of the cerebellum. Higher power images should be shown.”

There is no off-target “leaky” expression of Cre in MLIs of *Pcp2::Cre^{tdhu}* mice (Witter et al., 2016). This is why we chose to use this line in our investigation. The reviewer’s confusion stems from our poor choice of wording (“*this study*” was in reference to Nguyen-Vu et al., 2012, which we cited in the previous sentence). As mentioned below, we believe leaky Cre in MLIs is worrying and likely contributes to differences between our results and those of Nguyen-Vu et al. We again apologize and, as pointed out in our response to the first reviewer, **we edited the text to make this point clear (p. 16).**

“The explanation on page 15 of the different findings of this work and that of Nguyen-Vu, et al. needs expansion. How does this manuscript's finding of a lack of effect after ipsiversive training in the low-intensity condition compare to the author's suggestion that Nguyen-Vu et al. initiated LTP? What of the differences between this manuscript's and Nguyen-Vu et al.'s climbing fiber stimulation experiments?”

As stated in the text, when measuring the adaptive effect of optogenetic climbing fiber stimulation on VOR performance, we obtained the same result as Nguyen-Vu et al., 2012. Specifically, the pairing procedure resulted in an increase in VOR gain, relative to the control condition, when the optogenetic stimulus was timed to the ipsiversive phase of vestibular motion. We observed a similar adaptive effect on VOR performance when ChR2-expressing PCs were optogenetically activated at high intensity on the same phase of vestibular motion suggesting that convergent plasticity pathways were engaged. This makes sense because both types of stimuli evoke large-magnitude Ca^{2+} responses in PCs *in vivo*. Therefore the expectation is that they should substitute for each other which we confirmed. **We further clarified this point in the text to avoid any confusion (p. 16).**

Our results and those of Nguyen-Vu et al. differ in that they observed gain-increase VOR adaptation when pairing optogenetic-induced PC activity on the contraversive phase of vestibular motion whereas we observed gain-decrease learning. As we point out in the text, we surmise that these differences may be attributable to off-target stimulation of MLIs provided that the line used in their report (*Pcp2::Cre^{Mpin}*) has leaky expression of Cre in these interneurons and MLI activity is known to influence plasticity induction and VOR learning (Rowan et al. 2018). To help alleviate the reviewer's concerns regarding the discrepancies between our work and Nguyen-Vu et al., we obtained *Pcp2::Cre^{Mpin}* mice and crossed them with the Ai27 line to express ChR2 in Cre positive cells and performed the VOR optogenetic pairing experiment. We observed an identical result as Nguyen-Vu et al.: gain-increase learning resulted when the optogenetic stimulus was timed to the end of contraversive vestibular motion, relative to a control session of vestibular motion alone. This result conclusively shows that differences in the learning direction for contraversive pairing are attributable to the mouse strain used for driving Cre-mediated expression of ChR2. As the reviewer points out in their comment above, leaky Cre expression in MLIs is indeed worrying for the Nguyen-Vu et al. study. This is why we chose a line with specific expression of Cre in PCs.

We have decided to not include this new data in our manuscript because it would not bring new knowledge regarding the neurobiology of cerebellar learning. Rather, it would be perceived as invective against the Nguyen-Vu et al. study which we have been attempting to avoid. To gain full mechanistic insight into cerebellar learning, it would require further understanding as to why the sensitivity and direction of optogenetically induced VOR adaptation differs between the *Pcp2::Cre^{Mpin}* and *Pcp2::Cre^{Jdhu}* strains. This would require laborious measurements to compare optogenetically evoked Ca^{2+} responses in PCs of the *Pcp2::Cre^{Mpin}* strain and/or perform additional experiments in the *Pcp2::Cre^{Jdhu}* line to simultaneously manipulate the activity of both PCs and MLIs in an attempt to match the type of learning obtained in the *Pcp2::Cre^{Mpin}* mouse line. This strikes us as a lot of work for little gain. However, we agree that it is important to help readers understand the discrepancy between the results of this previous study and our own work. **Therefore, we have added additional text to the manuscript to help contextualize our findings with this previous literature (p. 17).**

“The authors are presenting a lot of experimental manipulations and results. It would be beneficial for readers to provide a summary table with all the results that allow for direct comparison between the different conditions. Alternatively, a summary figure with the proposed mechanism would be beneficial to readers that are less familiar with this field.”

We thank the reviewer for acknowledging that we have performed an enormous amount of work; our intent is to be thorough, which we believe we have accomplished. **Based on the reviewer's suggestion, we now include a summary cartoon depicting the proposed mechanism (Supplementary Fig. 5).** We believe this will help readers more easily integrate our results. **We also include a summary table including all comparisons.**

“For supplemental figure 4 and figure 4, why were the stimulation experiments only performed in one direction?”

Optogenetic-induced climbing fiber activation drives learning only when the stimulus occurs during the ipsiversive phase of vestibular motion (Nguyen-Vu et al., 2012; Kimpo et al., 2014; Rowan et al., 2018). For this reason, we only used ipsiversive pairing. Notably, relative to these previous publications, we shifted the timing of the stimulus to the end of the head turn so that it matched the timing for PC optogenetic activation. Regarding optogenetic PC pairing, we only tested contexts that we established were effective

for inducing learning (i.e., for contraversive pairing during low-intensity optogenetic stimuli and ipsiversive pairing for high-intensity optogenetic stimuli). **This is now stated in the text (p. 10).**

“Figure 5f: Please clarify what the control experiment is. And, can climbing fiber stimulation in mice injected with these viral constructs still induce LTD?”

For *ex vivo* plasticity experiments, the baseline period immediately prior to the pairing procedure serves as a reference to monitor for induced changes to synaptic strength after the pairing procedure. Either the pairing procedure produces a lasting change- an increase or decrease in synaptic strength as measured by the amplitude of evoked EPSPs- or it does not. **In response to the reviewer’s concern, we confirmed that pairing parallel fiber and climbing fiber activity results in LTD in the PCs of Kv2.1-ChR2 expressing mice. We include this new data in the manuscript (p. 12; Fig. 5f).**

“Please provide further discussion of what may be causing the increased Ca²⁺ when a greater light power is used. Do you believe this is activation of greater number of ChR2 channels or some other mechanism?”

Our recordings show that the size of the optogenetic-induced depolarization increases with light power. The increased size of the depolarization is likely due to the opening of additional ChR2 channels because their efficacy of opening increases in a light-dependent manner. We find that the resulting optogenetic-induced depolarization effectively opens voltage-gated Ca²⁺ channels resulting in Ca²⁺ influx. The effectiveness of Ca²⁺ channel opening increases in a voltage-dependent manner. Thus, large-amplitude optogenetic-induced depolarizations are more effective at opening Ca²⁺ channels and driving enhanced Ca²⁺ influx. **We explain this further in the Discussion (p. 14).**

“The authors mention on page 11 adding more optogenetic stimulation pulses in order to “increase the overall level of optogenetically induced excitation.” Please provide further detail in the text and figures to how many different levels of optogenetic stimulation were tried, the nature of the stimulus pattern, and the results of each attempt. How did the number and timing of spikes elicited in the soma-targeted stimulations compare to the whole-cell expression of ChR2?”

We performed the experiment using a single level of PC excitation, as measured by evoked action potential in the soma, because we knew, based on our Ca²⁺ imaging results, that somatically induced optogenetic excitation was rather ineffective at inducing dendritic Ca²⁺ elevation. Provided the well-known Ca²⁺ sensitivity of parallel-fiber-to-PC synaptic plasticity, it seemed logical to use a stimulus condition that would be more apt at inducing at least a modicum of dendritic Ca²⁺ entry. However, as presented in the results, this level of stimulation was ineffective at inducing a change in synaptic strength. **We attempted to clarify our language to more precisely indicate that a single intensity of stimulation was used (p. 11).**

The statement: “Over the time course of training, optogenetically induced simple spiking in PCs was insufficient to elicit behavioral change,” is a bit vague and confusing. Is this meant to only reference the soma-targeted ChR2 or was there generally no obvious change in behavior?”

It was meant that simple spiking did not instruct learning. This hypothesis was tested using soma-targeted ChR2, as this approach allowed us to activate PCs without evoking dendritic Ca²⁺ signals. In our statement referenced by the reviewer, we were being cautious to qualify that optogenetically evoked PC activation occurred only over the duration of the pairing procedure because it remains possible that induced spiking over much longer time periods could instruct learning. However, we did not test this possibility. **We changed our wording to make this clear (p. 14).**

“Successful targeting of optical fibers must be demonstrated.”

Optogenetic stimulation of floccular PCs drives eye movements in quiescent mice. Therefore, for each experimental mouse, we could confirm successful targeting of floccular PCs by the optical fiber implant using light-evoked eye movements. In rare cases where we failed to evoke eye movements, we removed the mice from the study and *post hoc* inspection confirmed misplaced targeting of the flocculus by the fiber optic implant. **We explained this further in the Methods section (p. 23).** We also routinely inspected optical fiber placement in mice with evoked eye movements to confirm the accuracy of our targeting procedure, however this was not done systematically.

“Please clarify where in the cerebellum the slice recordings and Ca²⁺ imaging of Purkinje cells were performed and justify why this location was chosen.”

As previously stated in the Methods section, we performed *ex vivo* recordings from PCs in slices obtained from the cerebellar vermis.

“Number of mice used per experiment/ information about experimental replication is lacking throughout the paper. Additionally, in the cases when “n” is present, it is not clearly defined.

We edited the manuscript to include more information about experimental replication including defining what “n” is referring to in a number of instances in the figures (e.g., PCs in slice experiments or mice in behavior experiments). We have also prepared a table (Supplementary Table 1) detailing a description of “n” for each experiment as well as the statistical test used for determining significance.

“For the ex vivo recordings, what does the n-number represent? Number of mice? Cells? If cells, from how many mice did these cells come?”

When reporting results from *ex vivo* recordings, “n” represents the number of cells. This is the standard reporting metric for this type of recording. **We have attempted to clarify this in the text and figures.** We also state that for each experimental condition, the number of mice used. As mentioned above, we have prepared a supplemental table that provides additional detail.

“I would recommend providing the mean and SEM for the statement: “the change in synaptic efficacy obtained by conjunctive pairing with climbing fibers was greater than that obtained by optogenetically induced PC activation (p = 0.04, t-test).”

The mean and SEM are already provided in the previous two sentences (i.e., EPSP amplitude 0.084 ± 0.04 and 0.64 ± 0.10 of baseline for conjunctive pairing with the optogenetic stimulus or climbing fibers, respectively).

“Please state the statistical values of panels 4e and 4f in the main body of the text.”

We now state the statistical values for Figure 4 in the text (p. 11).

“Methods: Can the authors please clarify when they used the different post-hoc tests for the ANOVA?”

We now state that we use Sidak, Tukey, or Dunnett’s post-hoc comparison tests, depending on whether comparing matched or unmatched values and whether comparing every mean to every other mean or comparing every mean to a control mean (p. 25). We also state the post hoc test used for each ANOVA test in the figure legends and provide a supplemental table with more detail.

“Figure 5b: the order of the example traces (left) is different from the order of the bars (right). For clarity and simplification, it would be beneficial to be consistent with the order.”

We thank the reviewer for this helpful comment. **We reorganized the example traces so that they match the order of the bar graph.**

“Figure 6d should be discussed before Figure 6f as it is presented as such in the figure (or the order of the text should be reversed).”

We reordered Fig. 6 so panels 6e and 6f were not presented before 6d in the Results section.

“The colors of the dotted lines in panels 3d and 3g are too faint to be appreciated. Please saturate the colors further.”

We saturated the colors of the dotted lines in Figure 3 to make them more distinct.

“Page 23: I would recommend reporting drug dosages in mg/kg.”

We now report drug doses as mg/kg (pp. 23 and 25).

“Is a period missing at the end of the sentence in figure legend for panel 6d or is there information that is cut off?”

This typo has been corrected (p. 38).

Reviewer #3 (Kamran):

“The manuscript by Bonnan et al is a tour de force examination of the Purkinje cell dendritic calcium signal amplitudes that drive bidirectional parallel fiber plasticity in the cerebellar cortex. The authors take advantage of optogenetics to drive changes in the dendritic calcium in Purkinje cells, and “calibrate” the stimulations to generate calcium signals which span from sub to suprathreshold in comparison to activation of climbing fibers. The authors show, quite convincingly, that the calcium rise in the dendrites of Purkinje cells can bidirectionally produce synaptic plasticity. Then they demonstrate that by titrating the dendritic calcium changes they can produce VOR gain changes. These changes are not brought about by a change in simple spike firing rate, but require changes in dendritic calcium because when ChRd expression was restricted to the soma of Purkinje cells it failed to induce plasticity.

Lastly, and for me perhaps a bit of distraction, the authors show that plasticity in vivo is dependent on endocannabinoid signaling. Overall, I think this is an outstanding paper. The manuscript represents a rigorous set of experiments that have been expertly conducted, and clarifies an important question in the field, and in my opinion rights many published wrongs. I frankly struggled to identify issues that I had concerns about. There are two exceptions to this.”

We are pleased that the reviewer found our work exciting and that it helps settle many controversies in the cerebellar field regarding the interpretation of optogenetic stimulation experiments. We also appreciate the commendation regarding the rigor of our approach.

“Megan Carey’s lab has presented a body of work which suggests that the CB1 receptors are not, as suggested before, required for learning in vivo if the KO mice are made to walk at the same pace as the wild type mice. That story is at odds with the results reported here.”

In our report, we establish that endocannabinoid signaling is engaged downstream of optogenetic PC activation and that this activity is sufficient to induce plasticity that drives adaptive behavioral responses that converge with learning induced by VOR performance errors. Despite our best attempt (see our response to the second reviewer), we could not determine whether CB1 receptor signaling is actually necessary for

error-driven VOR learning. Therefore, we do not believe that our results directly contradict the recently published findings by the Carey group (Albergaria et al. 2020) who found that CB1 receptors are dispensable for eye-blink conditioning, a cerebellar-dependent learning task. Importantly, the Albergaria study failed to fully exclude that developmental compensation occurs in the absence endocannabinoid signaling in CB1 receptor KO mice, leaving open the possibility that endocannabinoid signaling plays a role in cerebellar learning in wildtype mice. A more exhaustive study will be required in the future to directly test this question. **To address the reviewer's concern, we now cite Albergaria et al. 2020 and point out the key caveats in this study relative to our own work and that caution is warranted in the interpretation of our results provided the artificiality of optogenetic PC stimulation (p. 18).**

“The second minor concern is that the calcium signals seems to have gotten bigger when voltage gated Na channels were blocked with TTX (supl figure 1), and I cannot understand what the cause of this increase could be.”

After completing an additional set of experiments, we now confirm that blocking Na⁺ channels with TTX indeed results in larger dendritic Ca²⁺ transients in PCs, at least when evoked by suprathreshold optogenetic stimuli. We did not pursue a mechanistic explanation in experimental detail because we believe this result is tangential to our main findings. However, a previous report (Aubry et al. Morain, 1991) showed that TTX leads to alterations in internal Ca²⁺ levels through a mechanism involving the Na⁺/Ca²⁺ exchanger; this, in turn, promotes Purkinje cell dendritic excitability due to reduced SK2 channel activity which normally resists dendritic spiking. **We now cite this work and posit that this effect accounts for the observed increase in the ChR2-evoked Ca²⁺ signal size in TTX. We also performed another set of pharmacology experiments in slices to further show that the evoked dendritic Ca²⁺ response in the absence of TTX is also sensitive to voltage-gated Ca²⁺ channel blockers (p. 6; Supplementary Fig. 1c).**

Reviewer #1 (Remarks to the Author):

I thank the authors who altered their manuscript in response to my comments. I believe that this has significantly strengthened the paper, which I recommend should be published.

Alanna Watt

Reviewer #2 (Remarks to the Author):

The authors have submitted a substantially revised version of their paper. Each and every point and comment was addressed with much care. I want to thank the authors for providing such a complete and thorough set of responses. I enjoyed reading the responses as much as the paper. This is an outstanding piece of work. I have no further concerns.

Below are our responses to the reviewers' comments.

Reviewer #1:

I thank the authors who altered their manuscript in response to my comments. I believe that this has significantly strengthened the paper, which I recommend should be published.

We thank the reviewer for finding that our revisions significantly improved our paper and recommending our paper for publication.

Reviewer #2:

The authors have submitted a substantially revised version of their paper. Each and every point and comment was addressed with much care. I want to thank the authors for providing such a complete and thorough set of responses. I enjoyed reading the responses as much as the paper. This is an outstanding piece of work. I have no further concerns.

We thank the reviewer for their very positive comments on our revised manuscript.